# SUFFICIENT AND DISENTANGLED REPRESENTATION LEARNING

## ABSTRACT

We propose a novel representation learning approach called sufficient and disentangled representation learning (SDRL). With SDRL, we seek a data representation that maps the input data to a lower-dimensional space with two properties: sufficiency and disentanglement. First, the representation is sufficient in the sense that the original input data is conditionally independent of the response or label given the representation. Second, the representation is maximally disentangled with mutually independent components and is rotation invariant in distribution. We show that such a representation always exists under mild conditions on the input data distribution based on optimal transport theory. We formulate an objective function characterizing conditional independence and disentanglement. This objective function is then used to train a sufficient and disentangled representation with deep neural networks. We provide strong statistical guarantees for the learned representation by establishing an upper bound on the excess error of the objective function and show that it reaches the nonparametric minimax rate under mild conditions. We also validate the proposed method via numerical experiments and real data analysis.

## 1 INTRODUCTION

Representation learning is a fundamental problem in machine learning and artificial intelligence (Bengio et al., 2013). Certain deep neural networks are capable of learning effective data representation automatically and achieve impressive prediction results. For example, convolutional neural networks, which can encode the basic characteristics of visual observations directly into the network architecture, is able to learn effective representations of image data (LeCun et al., 1989). Such representations in turn can be subsequently used for constructing classifiers with outstanding performance. Convolutional neural networks learn data representation with a simple structure that captures the essential information through the convolution operator. However, in other application domains, optimizing the standard cross-entropy and least squares loss functions do not guarantee that the learned representations enjoy any desired properties (Alain & Bengio, 2016). Therefore, it is imperative to develop general principles and approaches for constructing effective representations for supervised learning.

There is a growing literature on representation learning in the context deep neural network modeling. Several authors studied the internal mechanism of supervised deep learning from the perspective of information theory (Tishby & Zaslavsky, 2015; Shwartz-Ziv & Tishby, 2017; Saxe et al., 2019), where they showed that training a deep neural network that optimizes the information bottleneck (Tishby et al., 2000) is a trade-off between the representation and prediction at each layer. To make the information bottleneck idea more practical, deep variational approximation of information bottleneck (VIB) is considered in Alemi et al. (2016). Information theoretic objectives describing conditional independence such as mutual information are utilized as loss functions to train a representation-learning function, i.e., an encoder in the unsupervised setting (Hjelm et al., 2018; Oord et al., 2018; Tschannen et al., 2019; Locatello et al., 2019; Srinivas et al., 2020). There are several interesting extensions of variational autoencoder (VAE) (Kingma & Welling, 2013) in the form of VAE plus a regularizer, including beta-VAE (Higgins et al., 2017), Annealed-VAE (Burgess et al., 2018), factor-VAE (Kim & Mnih, 2018), beta-TC-VAE (Chen et al., 2018), DIP-VAE (Kumar et al., 2018). The idea of using a latent variable model has also been used in adversarial auto-

encoders (AAE) (Makhzani et al., 2016) and Wasserstein auto-encoders (WAE) (Tolstikhin et al., 2018). However, these existing works focus on the unsupervised representation learning.

A challenge of supervised representation learning that distinguishes it from standard supervised learning is the difficulty in formulating a clear and simple objective function. In classification, the objective is clear, which is to minimize the number of misclassifications; in regression, a least squares criterion for model fitting error is usually used. In representation learning, the objective is different from the ultimate objective, which is typically learning a classifier or a regression function for prediction. How to establish a simple criterion for supervised presentation learning has remained an open question (Bengio et al., 2013).

We propose a sufficient and disentangled representation learning (SDRL) approach in the context of supervised learning. With SDRL, we seek a data representation with two characteristics: sufficiency and disentanglement. In the context of representation learning, sufficient means that a good representation should preserve all the information in the data about the supervised learning task. This is a basic requirement and a long-standing principle in statistics. This is closely related to the fundamental concept of sufficient statistics in parametric statistical models (Fisher, 1922). A sufficient representation can be naturally characterized by the conditional independence principle, which stipulates that, given the representation, the original input data does not contain any additional information about the response variable.

In addition to the basic sufficiency property, the representation should have a simple statistical structure. Disentangling is based on the general notion that some latent causes underlie data generation process: although the observed data are typically high-dimensional, complex and noisy, the underlying factors are low-dimensional, independent and have a relatively simple statistical structure. There is a range of definitions of disentangling (Higgins et al., 2018; Eastwood & Williams, 2018; Ridgeway & Mozer, 2018; Do & Tran, 2020). Several metrics have been proposed for the evaluation of disentangling. However, none of these definitions and metrics have been turned into empirical criterions and algorithms for learning disentangled representations. We adopt a simple definition of disentangling which defines a representation to be disentangled if its components are independent (Achille & Soatto, 2018). This definition requires the representation to be maximally disentangled in the sense that the total correlation is zero, where the total correlation is defined as the KL divergence between the joint distribution of $g(\mathbf{x})$ and the product of the marginal distributions of its components (Watanabe, 1960).

In the rest of the paper, we first discuss the motivation and the theoretical framework for learning a sufficient and disentangled representation map (SDRM). This framework leads to the formulation of an objective function based on the conditional independence principle and a metric for disentanglement and invariance adopted in this work. We estimate the target SDRM based on the sample version of the objective function using deep neural networks and develop an efficient algorithm for training the SDRM. We establish an upper error bound on the measure of conditional independence and disentanglement and show that it reaches the nonparametric minimax rate under mild regularity conditions. This result provides strong statistical guarantees for the proposed method. We validate the proposed SDRL via numerical experiments and real data examples.

## 2 SUFFICIENT AND DISENTANGLED REPRESENTATION

Consider a pair of random vectors $(\mathbf{x}, \mathbf{y}) \in \mathbb{R}^p \times \mathbb{R}^q$, where $\mathbf{x}$ is a vector of input variables and $\mathbf{y}$ is a vector of response variables or labels. Our goal is to find a sufficient and disentangled representation of $\mathbf{x}$.

**Sufficiency** We say that a measurable map $g : \mathbb{R}^p \to \mathbb{R}^d$ with $d \leq p$ is a sufficient representation of $\mathbf{x}$ if

$$\mathbf{y} \perp\!\!\!\perp \mathbf{x} | g(\mathbf{x}), \tag{1}$$

that is, $\mathbf{y}$ and $\mathbf{x}$ are conditionally independent given $g(\mathbf{x})$. This condition holds if and only if the conditional distribution of $\mathbf{y}$ given $\mathbf{x}$ and that of $\mathbf{y}$ given $g(\mathbf{x})$ are equal. Therefore, the information in $\mathbf{x}$ about $\mathbf{y}$ is completely encoded by $g(\mathbf{x})$. Such a $g$ always exists, since if we simply take $g(x) = x$, then (1) holds trivially. This formulation is a nonparametric generalization of the basic condition in sufficient dimension reduction (Li, 1991; Cook, 1998), where it is assumed $g(x) = B^T x$ with $B \in \mathbb{R}^{p \times d}$ belonging to the Stiefel manifold, i.e., $B^T B = I_d$.

Denote the class of sufficient representations satisfying (1) by

$$\mathcal{F} = \{ \boldsymbol{g} : \mathbb{R}^p \to \mathbb{R}^d, \boldsymbol{g} \text{ measurable and satisfies } \mathbf{y} \perp\!\!\!\perp \mathbf{x} | \boldsymbol{g}(\mathbf{x}) \}.$$

We refer to $\mathcal{F}$ as a Fisher class because of its close connection with the concept of sufficient statistics (Fisher, 1922; Cook, 2007). For an injective measurable transformation $T : \mathbb{R}^d \to \mathbb{R}^d$ and $\boldsymbol{g} \in \mathcal{F}$, $T \circ \boldsymbol{g}(\mathbf{x})$ is also sufficient by the basic property of conditional probability. Therefore, the Fisher class $\mathcal{F}$ is invariant in the sense that

$$T \circ \mathcal{F} = \mathcal{F}, \quad \text{provided } T \text{ is injective},$$

where $T \circ \mathcal{F} = \{T \circ \boldsymbol{g} : \boldsymbol{g} \in \mathcal{F}\}$. An important class of transformations is the class of affine transformations, $T \circ \boldsymbol{g} = \boldsymbol{A}\boldsymbol{g} + \boldsymbol{b}$, where $\boldsymbol{A}$ is a $d \times d$ nonsingular matrix and $\boldsymbol{b} \in \mathbb{R}^d$.

**Disentanglement** We focus on the disentangled representations among those that are sufficient. Therefore, we start from the functions of the input data that are sufficient representations in the Fisher class $\mathcal{F}$. For any sufficient and disentangled representation $g(\mathbf{x})$, let $\Sigma_g = \text{Var}(g(\mathbf{x}))$. Since the components of $g(\mathbf{x})$ are disentangled in the sense that they are independent, $\Sigma_g$ is a diagonal matrix, thus $\Sigma_g^{-1/2} g(\mathbf{x})$ also has independent components. Therefore, we can always rescale $g(\mathbf{x})$ such that it has identity covariance matrix. To further simplify the statistical structure of a representation $\boldsymbol{g}$, we also require it to be rotation invariant in distribution, that is, $\boldsymbol{Q}\boldsymbol{g}(\mathbf{x}) = \boldsymbol{g}(\mathbf{x})$ in distribution for any orthogonal matrix $\boldsymbol{Q} \in \mathbb{R}^{d \times d}$. The Fisher class $\mathcal{F}$ is rotation invariant in terms of conditional independence, but not all its members are rotation invariant in distribution. By the Maxwell characterization of the Gaussian distributions (Maxwell, 1860; Bartlett, 1934; Bryc, 1995; Gyenis, 2017), a random vector of dimension two or more with independent components is rotation invariant in distribution if and only if it is Gaussian with zero mean and a spherical covariance matrix. Therefore, after absorbing the scaling factor, for a sufficient representation map to be disentangled and rotation invariant, it is necessarily distributed as $N_d(\boldsymbol{0}, \boldsymbol{I}_d)$. Let $\mathcal{M}$ be the Maxwell class of functions $\boldsymbol{g} : \mathbb{R}^d \to \mathbb{R}^d$, where $g(\mathbf{x})$ is disentangled and rotation invariant in distribution. By the Maxwell characterization, we can write

$$\mathcal{M} = \{ \boldsymbol{g} : \mathbb{R}^p \to \mathbb{R}^d, \boldsymbol{g}(\mathbf{x}) \sim \mathcal{N}(\boldsymbol{0}, \boldsymbol{I}_d) \}. \tag{2}$$

Now our problem becomes that of finding a representation in $\mathcal{F} \cap \mathcal{M}$, the intersection of the Fisher class and the Maxwell class.

The first question to ask is whether such a representation exists. The following result from optimal transport theory provides an affirmative answer and guarantees that $\mathcal{F} \cap \mathcal{M}$ is nonempty under mild conditions (Brenier, 1991; McCann, 1995; Villani, 2008).

**Lemma 2.1.** *Let $\mu$ be a probability measure on $\mathbb{R}^d$. Suppose it has finite second moment and is absolutely continuous with respect to the standard Gaussian measure, denoted by $\gamma_d$. Then it admits a unique optimal transportation map $T : \mathbb{R}^d \to \mathbb{R}^d$ such that $T_{\#}\mu = \gamma_d \equiv \mathcal{N}(\boldsymbol{0}, \boldsymbol{I}_d)$, where $T_{\#}\mu$ denotes the pushforward distribution of $\mu$ under $T$. Moreover, $T$ is injective $\mu$-almost everywhere.*

Denote the law of a random vector $\mathbf{z}$ by $\mu_{\mathbf{z}}$. Lemma 2.1 implies that, for any $\boldsymbol{g} \in \mathcal{F}$ with $\mathbb{E}\|\boldsymbol{g}(\mathbf{x})\|^2 < \infty$ and $\mu_{\boldsymbol{g}(\mathbf{x})}$ absolutely continuous with respect to $\gamma_d$, there exists a map $T^*$ transforming the distribution of $\boldsymbol{g}(\mathbf{x})$ to $\mathcal{N}(\boldsymbol{0}, \boldsymbol{I}_d)$. Therefore, $R^* := T^* \circ \boldsymbol{g} \in \mathcal{F} \cap \mathcal{M}$, that is,

$$\mathbf{x} \perp\!\!\!\perp \mathbf{y} | R^*(\mathbf{x}) \quad \text{and} \quad R^*(\mathbf{x}) \sim \mathcal{N}(\boldsymbol{0}, \boldsymbol{I}_d), \tag{3}$$

i.e., $R^*$ is a sufficient and disentangled representation map (SDRM).

## 3 OBJECTIVE FUNCTION FOR SDRL

The above discussions lay the theoretical foundation for formulating an objective function that can be used for constructing a SDRM $R^*$ satisfying (3), or equivalently, $R^* \in \mathcal{F} \cap \mathcal{M}$.

Let $\mathcal{V}$ be a measure of dependence between random variables $\mathbf{x}$ and $\mathbf{y}$ with the following properties: (a) $\mathcal{V}[\mathbf{x}, \mathbf{y}] \geq 0$ with $\mathcal{V}[\mathbf{x}, \mathbf{y}] = 0$ if and only if $\mathbf{x} \perp\!\!\!\perp \mathbf{y}$; (b) $\mathcal{V}[\mathbf{x}, \mathbf{y}] \geq \mathcal{V}[R(\mathbf{x}), \mathbf{y}]$ for all measurable function $R$; (c) $\mathcal{V}[\mathbf{x}, \mathbf{y}] = \mathcal{V}[R^*(\mathbf{x}), \mathbf{y}]$ if and only if $R^* \in \mathcal{F}$. The properties (a)-(c) imply that

$$R^* \in \mathcal{F} \Leftrightarrow R^* \in \arg\max_R \mathcal{V}[R(\mathbf{x}), \mathbf{y}] = \arg\min_R \{-\mathcal{V}[R(\mathbf{x}), \mathbf{y}]\}.$$

We use a divergence measure $\mathbb{D}$ to quantify the difference between $\mu_{R(\mathbf{x})}$ and $\gamma_d$, as long as this measure satisfies the condition $\mathbb{D}(\mu_{R(\mathbf{x})}\|\gamma_d) \geq 0$ for all measurable function $R$ and $\mathbb{D}(\mu_{R(\mathbf{x})}\|\gamma_d) = 0$ if and only if $R \in \mathcal{M}$.

Then the problem of finding an $R^* \in \mathcal{F} \cap \mathcal{M}$ can be expressed as a constrained minimization problem:

$$\underset{R}{\arg\min} -\mathcal{V}[R(\mathbf{x}), \mathbf{y}] \text{ subject to } \mathbb{D}(\mu_{R(\mathbf{x})}\|\gamma_d) = 0.$$

Its Lagrangian form is

$$\mathcal{L}(R) = -\mathcal{V}[R(\mathbf{x}), \mathbf{y}] + \lambda\mathbb{D}(\mu_{R(\mathbf{x})}\|\gamma_d), \tag{4}$$

where $\lambda \geq 0$ is a tuning parameter. This parameter provides a balance between the sufficiency property and the disentanglement constraint. A small $\lambda$ leads to a representation with more emphasis on sufficiency, while a large $\lambda$ yields a representation with more emphasis on disentanglement. We show in Lemma 4.1 below that any $R^*$ satisfying (3) is a minimizer of $\mathcal{L}(R)$. Therefore, we can train a SDRM by minimizing the empirical version of $\mathcal{L}(R)$.

There are several options for $\mathcal{V}$ with the properties (a)-(c) described above. For example, we can take $\mathcal{V}$ to be the mutual information $\mathcal{V}[R(\mathbf{x}), \mathbf{y}] = \mathbf{I}(R(\mathbf{x}); \mathbf{y})$. However, in addition to the estimation of the SDRM $R$, this choice requires the estimation of the density ratio between $p(\mathbf{y}, R(\mathbf{x}))$ and $p(\mathbf{y})p(R(\mathbf{x}))$, which is not an easy task. We can also use the conditional covariance operators on reproducing kernel Hilbert spaces (Fukumizu et al., 2009). To be specific, in this work we use the distance covariance (Székely et al., 2007) of $\mathbf{y}$ and $R(\mathbf{x})$, which has an elegant $U$-statistic expression, does not involve additional unknown quantities and is easy to compute. For the divergnce measure of two distributions, we use the $f$-divergence (Ali & Silvey, 1966), which includes the KL-divergence as a special case.

## 4 LEARNING SUFFICIENT AND DISENTANGLED REPRESENTATION

We first describe some essentials about distance covariance and $f$-divergence.

**Distance covariance** We first recall the concept of distance covariance (Székely et al., 2007), which characterizes the dependence of two random variables.

Let $\mathrm{i}$ be the imaginary unit $(-1)^{1/2}$. For any $\boldsymbol{t} \in \mathbb{R}^d$ and $\boldsymbol{s} \in \mathbb{R}^m$, let $\psi_{\mathbf{z}}(\boldsymbol{t}) = \mathbb{E}[\exp^{\mathrm{i}\boldsymbol{t}^T\mathbf{z}}], \psi_{\mathbf{y}}(\boldsymbol{s}) = \mathbb{E}[\exp^{\mathrm{i}\boldsymbol{s}^T\mathbf{y}}]$, and $\psi_{\mathbf{z},\mathbf{y}}(\boldsymbol{t}, \boldsymbol{s}) = \mathbb{E}[\exp^{\mathrm{i}(\boldsymbol{t}^T\mathbf{z}+\boldsymbol{s}^T\mathbf{y})}]$ be the characteristic functions of random vectors $\mathbf{z} \in \mathbb{R}^d, \mathbf{y} \in \mathbb{R}^q$, and the pair $(\mathbf{z}, \mathbf{y})$, respectively. The squared distance covariance $\mathcal{V}[\mathbf{z}, \mathbf{y}]$ is defined as

$$\mathcal{V}[\mathbf{z}, \mathbf{y}] = \int_{\mathbb{R}^{d+m}} \frac{|\psi_{\mathbf{z},\mathbf{y}}(\boldsymbol{t}, \boldsymbol{s}) - \psi_{\mathbf{z}}(\boldsymbol{t})\psi_{\mathbf{y}}(\boldsymbol{s})|^2}{c_d c_m \|\boldsymbol{t}\|^{d+1} \|\boldsymbol{s}\|^{q+1}} \mathrm{d}\boldsymbol{t}\mathrm{d}\boldsymbol{s}, \text{ where } c_d = \frac{\pi^{(d+1)/2}}{\Gamma((d+1)/2)}.$$

Given $n$ i.i.d copies $\{\mathbf{z}_i, \mathbf{y}_i\}_{i=1}^n$ of $(\mathbf{z}, \mathbf{y})$, an unbiased estimator of $\mathcal{V}$ is the empirical distance covariance $\widehat{\mathcal{V}}_n$, which can be elegantly expressed as a $U$-statistic (Huo & Székely, 2016)

$$\widehat{\mathcal{V}}_n[\mathbf{z}, \mathbf{y}] = \frac{1}{C_n^4} \sum_{1 \leq i_1 < i_2 < i_3 < i_4 \leq n} h\left((\mathbf{z}_{i_1}, \mathbf{y}_{i_1}), \cdots, (\mathbf{z}_{i_4}, \mathbf{y}_{i_4})\right), \tag{5}$$

where $h$ is the kernel defined by

$$h\left((\boldsymbol{z}_1, \boldsymbol{y}_1), \ldots, (\boldsymbol{z}_4, \boldsymbol{y}_4)\right) = \tfrac{1}{4} \sum_{\substack{1 \leq i,j \leq 4 \\ i \neq j}} \|\boldsymbol{z}_i - \boldsymbol{z}_j\| \|\boldsymbol{y}_i - \boldsymbol{y}_j\|$$

$$-\tfrac{1}{4} \sum_{i=1}^4 \left(\sum_{\substack{1 \leq j \leq 4 \\ j \neq i}} \|\boldsymbol{z}_i - \boldsymbol{z}_j\| \sum_{\substack{1 \leq j \leq 4 \\ i \neq j}} \|\boldsymbol{y}_i - \boldsymbol{y}_j\|\right) + \tfrac{1}{24} \sum_{\substack{1 \leq i,j \leq 4 \\ i \neq j}} \|\boldsymbol{z}_i - \boldsymbol{z}_j\| \sum_{\substack{1 \leq i,j \leq 4 \\ i \neq j}} \|\boldsymbol{y}_i - \boldsymbol{y}_j\|.$$

**f-divergence** Let $\mu$ and $\gamma$ be two probability measures on $\mathbb{R}^d$. The $f$-divergence (Ali & Silvey, 1966) between $\mu$ and $\gamma$ with $\mu \ll \gamma$ is defined as $\mathbb{D}_f(\mu\|\gamma) = \int_{\mathbb{R}^d} f(\frac{\mathrm{d}\mu}{\mathrm{d}\gamma})\mathrm{d}\gamma$, where $f : \mathbb{R}^+ \to \mathbb{R}$ is a differentiable convex function satisfying $f(1) = 0$. Let $f^*$ be the Fenchel conjugate of $f$ (Rockafellar, 1970), defined as $f^*(t) = \sup_{x \in \mathbb{R}}\{tx - f(x)\}, t \in \mathbb{R}$. The $f$-divergence admits the following variational formulation (Keziou, 2003; Nguyen et al., 2010; Nowozin et al., 2016).

**Lemma 4.1.**

$$\mathbb{D}_f(\mu\|\gamma) = \max_{D:\mathbb{R}^d \to \mathrm{dom}(f^*)} \mathbb{E}_{\mathbf{z}\sim\mu}[D(\mathbf{z})] - \mathbb{E}_{\mathbf{w}\sim\gamma}[f^*(D(\mathbf{w}))], \tag{6}$$

*where the maximum is attained at $D(\boldsymbol{z}) = f'(\frac{\mathrm{d}\mu}{\mathrm{d}\gamma}(\boldsymbol{z}))$.*

Commonly used divergence measures include the Kullback-Leibler (KL) divergence, the Jensen-Shanon (JS) divergence and the $\chi^2$-divergence.

**Learning SDRM** We are now ready to formulate an empirical objective function for learning SDR-M $R^*$. Let $R \in \mathcal{M}$, where $\mathcal{M}$ is the Maxwell class defined in (2). By the variational formulation (6), we can write the population version of the objective function (4) as

$$\mathcal{L}(R) = -\mathcal{V}[R(\mathbf{x}), \mathbf{y}] + \lambda \max_{D:\mathbb{R}^d \to \mathrm{dom}(f^*)} \{\mathbb{E}_{\mathbf{x} \sim \mu_{\mathbf{x}}}[D(R(\mathbf{x}))] - \mathbb{E}_{\mathbf{w} \sim \gamma_d}[f^*(D(\mathbf{w}))]\}. \quad (7)$$

This expression is convenient since we can simply replace the expectations by the corresponding empirical averages.

**Theorem 4.2.** *We have $R^* \in \arg\min_{R \in \mathcal{M}} \mathcal{L}(R)$ provided (3) holds.*

According to Theorem 4.2, it is natural to estimate $R^*$ based on the empirical version of the objective function (7) when a random sample $\{(\mathbf{x}_i, \mathbf{y}_i)\}_{i=1}^n$ is available. We estimate $R^*$ using deep neural networks. We employ two networks as follows:

- Representer network $R_{\boldsymbol{\theta}}$: This network is used for training $R^*$. Let $\mathcal{R}$ be the set of such neural networks $R_{\boldsymbol{\theta}} : \mathbb{R}^p \to \mathbb{R}^d$.

- Discriminator network $D_{\boldsymbol{\phi}}$: This network is used as the witness function for checking whether the distribution of the estimator of $R^*$ is approximately the same as $\mathcal{N}(\mathbf{0}, \boldsymbol{I}_d)$. Similarly, denote $\mathcal{D}$ as the set of such neural networks $D_{\boldsymbol{\phi}} : \mathbb{R}^d \to \mathbb{R}$.

Let $\{\mathbf{w}_i\}_{i=1}^n$ be $n$ i.i.d random vectors drawn from $\gamma_d$. The estimated SDRM is defined by

$$\widehat{R}_{\boldsymbol{\theta}} \in \arg\min_{R_{\boldsymbol{\theta}} \in \mathcal{R}} \widehat{\mathcal{L}}(R_{\boldsymbol{\theta}}) = -\widehat{\mathcal{V}}_n[R_{\boldsymbol{\theta}}(\mathbf{x}), \mathbf{y}] + \lambda \widehat{\mathbb{D}}_f(\mu_{R_{\boldsymbol{\theta}}(\mathbf{x})} \| \gamma_d), \quad (8)$$

where $\widehat{\mathcal{V}}_n[R_{\boldsymbol{\theta}}(\mathbf{x}), \mathbf{y}]$ is an unbiased and consistent estimator of $\mathcal{V}[R_{\boldsymbol{\theta}}(\mathbf{x}), \mathbf{y}]$ as defined in (5) based on $\{(R_{\boldsymbol{\theta}}(\mathbf{x}_i), \mathbf{y}_i), i = 1, \dots, n\}$ and

$$\widehat{\mathbb{D}}_f(\mu_{R_{\boldsymbol{\theta}}(\mathbf{x})} \| \gamma_d) = \max_{D_{\boldsymbol{\phi}} \in \mathcal{D}} \frac{1}{n} \sum_{i=1}^n [D_{\boldsymbol{\phi}}(R_{\boldsymbol{\theta}}(\mathbf{x}_i)) - f^*(D_{\boldsymbol{\phi}}(\mathbf{w}_i))]. \quad (9)$$

**Statistical guarantee** Since a SDRM $R^*$ is only identifiable up to orthogonal transforms under the constraint that $R^*(\mathbf{x}) \sim \mathcal{N}(0, \mathbf{I}_d)$, no consistency results for $\widehat{R}_{\boldsymbol{\theta}}$ itself can be obtained. But this is not a flaw of the proposed method. Indeed, the most important statistical guarantee of the learned $R^*$ is that the objective of conditional independence and disentanglement is achieved. Therefore, we establish an upper bound on the excess risk $\mathcal{L}(\widehat{R}_{\boldsymbol{\theta}}) - \mathcal{L}(R^*)$ of the deep nonparametric estimator $\widehat{R}_{\boldsymbol{\theta}}$ in (8). We make the following assumptions.

(A1) For any $\varepsilon > 0$, there is a constant $B_1 > 0$ such that $\mu_{\mathbf{x}}([-B_1, B_1]^p) > 1 - \varepsilon$, and $R^*$ is Lipschitz continuous on $[-B_1, B_1]^p$ with Lipschitz constant $L_1$.

(A2) For $R \in \mathcal{M}$, we assume $r(\boldsymbol{z}) = \frac{\mathrm{d}\mu_{R(\mathbf{x})}}{\mathrm{d}\gamma_d}(\boldsymbol{z})$ is Lipschitz continuous on $[-B_1, B_1]^p$ with Lipschitz constant $L_2$, and $0 < c_1 \le r(\boldsymbol{z}) \le c_2$.

Denote $B_2 = \max\{|f'(c_1)|, |f'(c_2)|\}$, $B_3 = \max_{|s| \le 2L_2\sqrt{d}\log n + B_2} |f^*(s)|$.

The specifications of the network parameters, including depth, width, size and the supremum norm over the domains of the representer $R_{\boldsymbol{\theta}}$ and the discriminator $D_{\boldsymbol{\phi}}$ are given in Appendix B.

**Theorem 4.3.** *Suppose $\lambda > 0$ and $\lambda = \mathcal{O}(1)$. Suppose conditions (A1)-(A2) hold and set the network parameters according to (i)-(ii). Then*

$$\mathbb{E}_{\{\mathbf{x}_i, \mathbf{y}_i, \mathbf{w}_i\}_{i=1}^n}[\mathcal{L}(\widehat{R}_{\boldsymbol{\theta}}) - \mathcal{L}(R^*)] \le C((L_1 + L_2)\sqrt{d}pn^{-\frac{2}{2+p}} + L_2\sqrt{d}(\log n)n^{-\frac{2}{2+d}}),$$

*where $C$ is a constant that depends on $B_1$, $B_2$ and $B_3$ but not on $n, q, p$ and $d$.*

The proof of this theorem is given in Appendix B. The result established in Theorem 4.3 provides strong statistical guarantees for the proposed method. The rate $n^{-2/(2+p)}$ matches the minimax non-parametric estimation rate for Lipschitz class contained in $\mathbb{R}^p$ of functions (Stone, 1982; Tsybakov, 2008). Up to a $\log n$ factor, the rate $(\log n)n^{-2/(2+d)}$ matches the minimax rate of nonparametric estimation of Lipschitz densities via GANs (Singh et al., 2018; Liang, 2018).

## 5 COMPUTATION

We can update $\boldsymbol{\theta}$ and $\boldsymbol{\phi}$ alternately as in training GANs (Goodfellow et al., 2014). However, this approach suffers from the instability issues. In our implementation, we utilize the more stable particle method based on gradient flow (Gao et al., 2019; 2020). The key idea is to find a sequence of nonlinear but simpler residual maps, say $\mathbb{T}(\boldsymbol{z}) = \boldsymbol{z} + s\mathbf{v}(\boldsymbol{z})$, pushing the samples from $\mu_{R_{\boldsymbol{\theta}}(\mathbf{x})}$ to the target distribution $\gamma_d$ along a velocity fields $\mathbf{v}(\boldsymbol{z}) = -\nabla f'(r(\boldsymbol{z}))$ that most decreases the $f$-divergence $\mathbb{D}_f(\cdot||\gamma_d)$ at $\mu_{R_{\boldsymbol{\theta}}(\mathbf{x})}$. The residual maps can be estimated via deep density-ratio estimators, which take the form $\mathbb{T}(\boldsymbol{z}) = \boldsymbol{z} + s\widehat{\mathbf{v}}(\boldsymbol{z}), \boldsymbol{z} \in \mathbb{R}^d$, where $s$ is a step size and $\widehat{\mathbf{v}}(\boldsymbol{z}) = -f''(\hat{r}(\boldsymbol{z}))\nabla\hat{r}(\boldsymbol{z})$. Here $\hat{r}(\boldsymbol{z})$ is an estimated density ratio of the density of $R_{\boldsymbol{\theta}}(\mathbf{x})$ at the current value of $\boldsymbol{\theta}$ over the density of the reference distribution. We use $\mathbb{T}$ to transform $\mathbf{z}_i = R_{\boldsymbol{\theta}}(\mathbf{x}_i), i = 1, \ldots, n$ into Gaussian samples. Once this is done, we update $\boldsymbol{\theta}$ via minimizing the loss $-\widehat{\mathcal{V}}_n[R_{\boldsymbol{\theta}}(\mathbf{x}), \mathbf{y}] + \lambda \sum_{i=1}^n \|R_{\boldsymbol{\theta}}(\mathbf{x}_i) - \mathbf{z}_i\|^2/n$. We describe the algorithm below.

- Input $\{\mathbf{x}_i, \mathbf{y}_i\}_{i=1}^n$. Tuning parameters: $s, \lambda, d$. Sample $\{\mathbf{w}_i\}_{i=1}^n \sim \gamma_d$.
- **Outer loop for $\boldsymbol{\theta}$**
  - **Inner loop (particle method)**
    * Let $\mathbf{z}_i = R_{\boldsymbol{\theta}}(\mathbf{x}_i), i = 1, 2, ..., n$.
    * Solve $\widehat{D}_{\boldsymbol{\phi}} \in \arg\min_{Q_{\boldsymbol{\phi}}} \sum_{i=1}^n \frac{1}{n} \left(\log(1 + \exp^{D_{\boldsymbol{\phi}}(\mathbf{z}_i)}) + \log(1 + \exp^{-D_{\boldsymbol{\phi}}(\mathbf{w}_i)})\right)$.
    * Define the residual map $\mathbb{T}(\boldsymbol{z}) = \boldsymbol{z} - sf''(\hat{r}(\boldsymbol{z}))\nabla\hat{r}(\boldsymbol{z})$ with $\hat{r}(\boldsymbol{z}) = \exp^{-\widehat{D}_{\boldsymbol{\phi}}(\boldsymbol{z})}$.
    * Update the particles $\mathbf{z}_i = \mathbb{T}(\mathbf{z}_i), i = 1, 2, ..., n$.
  - **End inner loop**
  - Update $\boldsymbol{\theta}$ via minimizing $-\widehat{\mathcal{V}}_n[R_{\boldsymbol{\theta}}(\mathbf{x}), \mathbf{y}] + \lambda \sum_{i=1}^n \|R_{\boldsymbol{\theta}}(\mathbf{x}_i) - \mathbf{z}_i\|^2/n$ using SGD.
- **End outer loop**

## 6 EXPERIMENTS

We evaluate the proposed SDRL with the KL-divergence using both simulated and real data. The goal of our experiments is to demonstrate that the representations trained based the proposed method perform well. Our proposed method is not trying to learn a classifier or a regression function directly, but rather to learn representation that preserve all the information. So our experiments are designed to evaluate the performance of simple classification and regression methods using the representations we learned as input. The results demonstrate that a simple classification or regression model using the representations we trained performs better than or comparably with the best classification or regression method using deep neural networks.

Details on the network structures and hyperparameters are included in Appendix A. Our experiments were conducted on Nvidia DGX Station workstation using a single Tesla V100 GPU unit. The PyTorch code of SDRL is available at `https://github.com/anonymous/SDRL`.

### 6.1 SIMULATED DATA

In this subsection, we evaluate SDRL on simulated regression and classification problems.

**Regression** We generate $5,000$ data points from two models. Model A: $\mathbf{y} = \mathbf{x}_1[0.5 + (\mathbf{x}_2 + 1.5)^2]^{-1} + (1 + \mathbf{x}_2)^2 + \sigma\boldsymbol{\varepsilon}$, where $\mathbf{x} \sim N(\mathbf{0}, \boldsymbol{I}_4)$; Model B: $\mathbf{y} = \sin^2(\pi\mathbf{x}_1 + 1) + \sigma\boldsymbol{\varepsilon}$, where $\mathbf{x} \sim \text{Uniform}([0, 1]^4)$. In both models, $\boldsymbol{\varepsilon} \sim N(\mathbf{0}, \boldsymbol{I}_4)$. We use a 3-layer network with ReLU activation for $R_{\boldsymbol{\theta}}$ and a single hidden layer ReLU network for $D_{\boldsymbol{\phi}}$. We compare SDRL with two prominent sufficient dimension reduction methods: sliced inverse regression (SIR) (Li, 1991) and sliced average variance estimation (SAVE) (Cook & Weisberg, 1991). We fit a linear model with the learned features and the response variable, and report the prediction errors in Table 1. We see that SDRL outperforms SIR and SAVE in terms of prediction error.

**Classification** We visualize the learned features of SDRL on two simulated datasets. We first generate (1) 2-dimensional concentric circles from two classes as in Figure 1 (a); (2) 2-dimensional moons data from two classes as in Figure 1 (e); (3) 3-dimensional Gaussian data from six classes

Table 1: Averaged prediction errors and their standard errors (based on 5-fold validation).

| | Model A | | | Model B | | |
|---|---|---|---|---|---|---|
| Method | $\sigma = 0.1$ | $\sigma = 0.4$ | $\sigma = 0.8$ | $\sigma = 0.1$ | $\sigma = 0.2$ | $\sigma = 0.3$ |
| SDRL | **1.101 ± .193** | **1.179 ± .117** | **1.401 ± .159** | **0.149 ± .050** | **0.231 ± .025** | **0.325 ± .026** |
| SIR | 1.521 ± .133 | 1.614 ± .223 | 1.704 ± .095 | 0.266 ± .003 | 0.319 ± .004 | 0.391 ± .010 |
| SAVE | 1.521 ± .134 | 1.614 ± .221 | 1.702 ± .098 | 0.266 ± .003 | 0.319 ± .004 | 0.391 ± .010 |

as in Figure 1 (i). In each dataset, we generate 5,000 data points for each class. We next map the data into 100-dimensional space using matrices with entries i.i.d Unifrom($[0, 1]$). Finally, we apply SDRL to these 100-dimensional datasets to learn 2-dimensional features. We use a 10-layer dense convolutional network (DenseNet) (Huang et al., 2017) as $R_{\theta}$ and a 4-layer network as $D_{\phi}$. We display the evolutions of the learned 2-dimensional features by SDRL in Figure 1. For ease of visualization, we push all the distributions onto the uniform distribution on the unit circle, which is done by normalizing the standard Gaussian random vectors to length one. Clearly, the learned features for different classes in the examples are well disentangled.

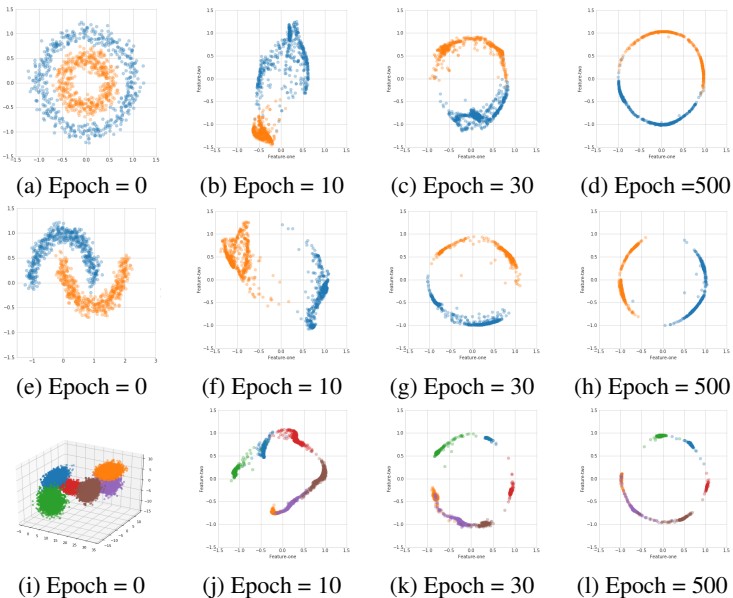

| (a) Epoch = 0 | (b) Epoch = 10 | (c) Epoch = 30 | (d) Epoch =500 |
|---|---|---|---|

| (e) Epoch = 0 | (f) Epoch = 10 | (g) Epoch = 30 | (h) Epoch = 500 |
|---|---|---|---|

| (i) Epoch = 0 | (j) Epoch = 10 | (k) Epoch = 30 | (l) Epoch = 500 |
|---|---|---|---|

Figure 1: Evolving learned features. The first, second and third rows show concentric circles, moons and 3D Gaussian datasets, respectively.

## 6.2 REAL DATASETS

**Regression** We use a benchmark YearPredictionMSD dataset to demonstrate the prediction performance of SDRL (https://archive.ics.uci.edu/ml/datasets/YearPredictionMSD). This dataset has 515,345 observations with 90 predictors. The problem is to predict the year of song release. We randomly split the data into five parts for cross validated evaluation of the prediction performance. We employ a 3-layer network for both $D_{\phi}$ and $R_{\theta}$. A linear regression model is fitted using the learned representations and the response. The mean prediction errors and their standard errors based on SDRL, principal component analysis (PCA), sparse principal component analysis (SPCA) and ordinary least squares (OLS) regression with original data are reported in Table 2. SDRL outperforms PCA, SPCA and OLS in terms prediction accuracy.

**Classification** We benchmark the classification performance of SDRL using MNIST (LeCun et al., 2010), FashionMNIST (Xiao et al., 2017), and CIFAR-10 (Krizhevsky et al., 2009) against alterna-

Table 2: Prediction error $\pm$ standard error: YearPredictionMSD dataset

| Methods | $d = 10$ | $d = 20$ | $d = 30$ | $d = 40$ |
|---|---|---|---|---|
| SDRL | $\mathbf{8.8 \pm 0.1}$ | $\mathbf{9.2 \pm 0.8}$ | $\mathbf{9.2 \pm 0.8}$ | $\mathbf{8.8 \pm 0.1}$ |
| SPCA | $10.6 \pm 0.1$ | $10.4 \pm 0.1$ | $9.6 \pm 0.1$ | $10.2 \pm 0.1$ |
| PCA | $10.6 \pm 0.1$ | $10.4 \pm 0.1$ | $10.3 \pm 0.1$ | $10.2 \pm 0.1$ |
| OLS | | ———9.6 $\pm$0.1——— | | |

tive methods including convolutional networks (CN) and distance correlation autoencoder (dCorAE) (Wang et al., 2018). With CN, we use the feature extractor by dropping the cross entropy layer of the DenseNet trained for classification. The MNIST and FashionMNIST datasets consist of 60k and 10k grayscale images with $28 \times 28$ pixels for training and testing, respectively, while the CIFAR-10 dataset contains 50k and 10k colored images with $32 \times 32$ pixels for training and testing, respectively. For the learning from scratch strategy, the represener network $R_{\boldsymbol{\theta}}$ has 20 layers for MNIST data and 100 layers for CIFAR-10 data. We apply the transfer learning technique to the combination of SDRL and CN on CIFAR-10 (Krizhevsky et al., 2009). The pretrained WideResnet-101 model (Zagoruyko & Komodakis, 2016) on the Imagenet dataset with Spinal FC (Kabir et al., 2020) is adopt for $R_{\boldsymbol{\theta}}$. The discriminator network $D_{\boldsymbol{\phi}}$ is a 4-layer network. The the architecture of $R_{\boldsymbol{\theta}}$ and most hyperparameters are shared across all four methods - SDRL, CN, SDRL+CN and dCorAE. Finally, we use the $k$-nearest neighbor ($k = 5$) classifier on the learned features for all methods.

The classification accuracies are reported in Tables 3 and 4. We can see that the classification accuracies of SDRL are comparable with those of CN and dCorAE. As shown in Table 4, the classification accuracies of CN leveraging SDRL outperforms those of CN. We also calculate the estimated distance correlation (DC) between the learned features and their labels as $\rho_{\mathbf{z},\mathbf{y}}^2 = \mathcal{V}[\mathbf{z},\mathbf{y}]^2 / \sqrt{(\mathcal{V}[\mathbf{z}]^2 \times \mathcal{V}[\mathbf{y}]^2)}$, where $\mathcal{V}[\mathbf{z}]$ and $\mathcal{V}[\mathbf{y}]$ are the distance variances such that $\mathcal{V}[\mathbf{z}] = \mathcal{V}[\mathbf{z},\mathbf{z}]$, $\mathcal{V}[\mathbf{y}] = \mathcal{V}[\mathbf{y},\mathbf{y}]$. For more details, please see Székely et al. (2007). Figure 2 shows the DC values MNIST, FashionMNIST and CIFAR-10 data. SDRL and SDRL+CN achieves higher DC values.

Table 3: Classification accuracy for MNIST and FashionMNIST

| | MNIST | | | FashionMNIST | | |
|---|---|---|---|---|---|---|
| $d$ | SDRL | dCorAE | CN | SDRL | dCorAE | CN |
| $d = 16$ | 99.41 | 99.58 | 99.39 | **94.44** | 94.18 | 94.21 |
| $d = 32$ | **99.61** | 99.54 | 99.45 | 94.18 | 93.89 | 94.41 |
| $d = 64$ | **99.56** | 99.53 | 99.49 | 94.13 | 94.24 | 94.38 |

Table 4: Classification accuracy for CIFAR-10 data

| | Learning from scratch | | | Transfer learning | | |
|---|---|---|---|---|---|---|
| $d$ | SDRL | dCorAE | CN | SDRL | CN | SDRL+CN |
| $d = 16$ | **94.29** | 94.15 | 94.21 | 97.52 | 97.44 | **97.68** |
| $d = 32$ | 94.58 | 94.18 | 94.92 | 97.33 | 97.79 | **97.96** |
| $d = 64$ | 94.46 | 94.66 | 95.09 | 97.49 | 97.90 | **97.91** |

## 7 CONCLUSION AND FUTURE WORK

In this work, we formulate a framework for sufficient and disentangled representation learning and construct an objective function characterizing conditional independence and disentanglement. This enables us to learn a representation with the desired properties empirically. We provide statistical guarantees for the learned representation by deriving an upper bound on the excess risk of the objective function.

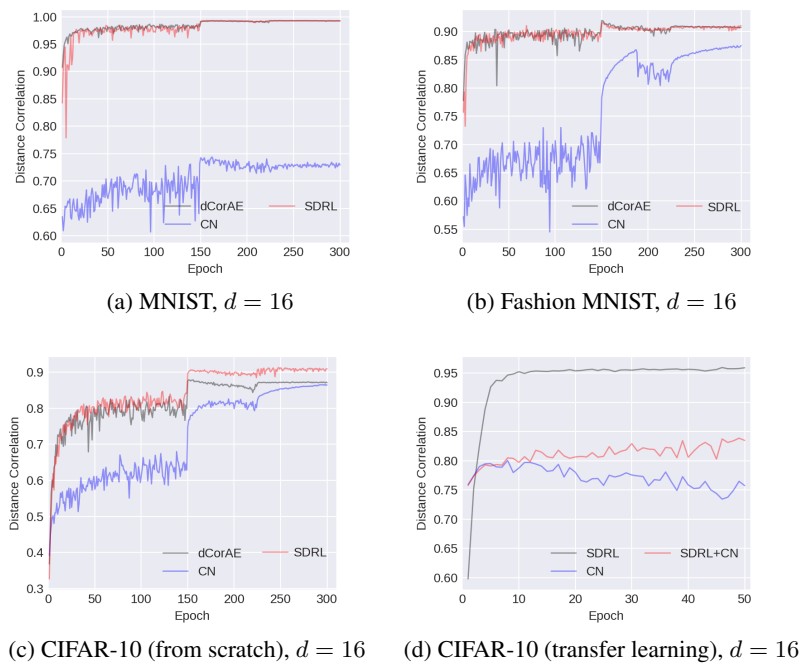

(a) MNIST, $d = 16$  (b) Fashion MNIST, $d = 16$

(c) CIFAR-10 (from scratch), $d = 16$  (d) CIFAR-10 (transfer learning), $d = 16$

Figure 2: The distance correlations of labels with learned features based on SDRL, CN, SDRL+CN and dCorAE for FashionMNIST and CIFAR-10 data.

There are several questions that deserve further study. First, we can adopt different measures of conditional independence including mutual information and conditional covariance operators on reproducing kernel Hilbert spaces (Fukumizu et al., 2009). We can also use other divergence measures such as the Wasserstein distance in the objective function. Finally, Lemma 2.1 suggests that the intersection of the Fisher class $\mathcal{F}$ and the Maxwell class $\mathcal{M}$ can still be large, and there can be many statistically equivalent representations in $\mathcal{F} \cap \mathcal{M}$. We can make further reduction of $\mathcal{F} \cap \mathcal{M}$ by imposing additional constraints, for example, certain minimal properties, sparsity, and robustness against noise perturbation.

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

## A   APPENDIX: EXPERIMENTAL DETAILS

### A.1   SIMULATION STUDIES

The values of the hyper-parameters for the simulated experiments are given in Table A1, where $\lambda$ is the penalty parameter, $d$ is the dimension of the SDRM, $n$ is the mini-batch size in SGD, $T_1$ is the number of inner loops to push forward particles $\mathbf{z}_i$, $T_2$ is the number of outer loops for training $R_{\boldsymbol{\theta}}$, and $s$ is the step size to update particles. For the regression models, the neural network architectures are shown in Table A2

As shown in Table A3, a multilayer perceptron (MLP) is utilized for the neural structure $D_{\boldsymbol{\phi}}$ in the classification problem. The detailed architecture of 10-layer dense convolutional network (DenseNet) (Huang et al., 2017; Amos & Kolter) deployed for $R_{\boldsymbol{\theta}}$ is shown in Table A4. For all the settings, we adopted the Adam (Kingma & Ba, 2014) optimizer with an initial learning rate of 0.001 and weight decay of 0.0001.

Table A1: Hyper-parameters for simulated examples, where $s$ varies according to epoch

| Task | $\lambda$ | $d$ | $n$ | $T_1$ | $T_2$ | 0-150 | 151-225 | 226-500 |
|---|---|---|---|---|---|---|---|---|
| | | | | | | | $s$ | |
| Regression | 1.0 | 2 or 1 | 64 | 1 | 500 | 3.0 | 2.0 | 1.0 |
| Classification | 1.0 | 2 | 64 | 1 | 500 | 2.0 | 1.5 | 1.0 |

Table A2: MLP architectures for $D_\phi$ and $R_\theta$ in regression

| | $D_\phi$ | | | $R_\theta$ | |
|---|---|---|---|---|---|
| Layers | Details | Output size | | Details | Output size |
| Layer 1 | Linear, LeakyReLU | 16 | | Linear, LeakyReLU | 16 |
| Layer 2 | Linear | 1 | | Linear, LeakyReLU | 8 |
| Layer 3 | | | | Linear | $d$ |

Table A3: MLP architecture for $D_\phi$ of simulated classification examples and the benchmark classification datasets

| Layers | Details | Output size |
|---|---|---|
| Layer 1 | Linear, LeakyReLU | 64 |
| Layer 2 | Linear, LeakyReLU | 128 |
| Layer 3 | Linear, LeakyReLU | 64 |
| Layer 4 | Linear | 1 |

Table A4: DenseNet architecture for $R_\theta$ in the simulated classification examples

| Layers | Details | Output size |
|---|---|---|
| Convolution | $3 \times 3$ Conv | $24 \times 20 \times 20$ |
| Dense Block 1 | $\begin{bmatrix} \text{BN, } 1 \times 1 \text{ Conv} \\ \text{BN, } 3 \times 3 \text{ Conv} \end{bmatrix} \times 1$ | $36 \times 20 \times 20$ |
| Transition Layer 1 | BN, ReLU, $2 \times 2$ Average Pool, $1 \times 1$ Conv | $30 \times 10 \times 10$ |
| Dense Block 2 | $\begin{bmatrix} \text{BN, } 1 \times 1 \text{ Conv} \\ \text{BN, } 3 \times 3 \text{ Conv} \end{bmatrix} \times 1$ | $18 \times 10 \times 10$ |
| Transition Layer 2 | BN, ReLU, $2 \times 2$ Average Pool, $1 \times 1$ Conv | $15 \times 5 \times 5$ |
| Dense Block 3 | $\begin{bmatrix} \text{BN, } 1 \times 1 \text{ Conv} \\ \text{BN, } 3 \times 3 \text{ Conv} \end{bmatrix} \times 1$ | $27 \times 5 \times 5$ |
| Pooling | BN, ReLU, $5 \times 5$ Average Pool, Reshape | 27 |
| Fully connected | Linear | 2 |

## A.2 REAL DATASETS

**Regression:** In the regression problems, hyper-parameters are presented in Table A5. The Adam optimizer with an initial learning rate of 0.001 and weight decay of 0.0001 is adopted. The MLP architectures of $D_\phi$ and $R_\theta$ for the YearPredictionMSD data are shown in Table A6.

Table A5: Hyper-parameters for YearPredictionMSD data

| Dataset | $\lambda$ | $d$ | $n$ | $T_1$ | $T_2$ | $s$ |
|---|---|---|---|---|---|---|
| YearPredictionMSD | 1.0 | 10, 20, 30, 40 | 64 | 1 | 500 | 1.0 |

Table A6: MLP architectures for $D_\phi$ and $R_\theta$ for YearPredictionMSD data

| | $D_\phi$ | | | $R_\theta$ | |
|---|---|---|---|---|---|
| Layers | Details | Output size | | Details | Output size |
| Layer 1 | Linear, LeakyReLU | 32 | | Linear, LeakyReLU | 32 |
| Layer 2 | Linear, LeakyReLU | 8 | | Linear, LeakyReLU | 8 |
| Layer 3 | Linear | 1 | | Linear | $d$ |

**Classification:** For the classification problems, hyper-parameters are shown in Table A7. We again use Adam as the SGD optimizers for both $D_\phi$ and $R_\theta$. Specifically, learning rate of 0.001 and weight decay of 0.0001 are used for $D_\phi$ in all datasets and for $R_\theta$ on MNIST (LeCun et al., 2010). We customized the SGD optimizers with momentum at 0.9, weight decay at 0.0001, and learning rate $\rho$ in Table A8 for FashionMNIST (Xiao et al., 2017) and CIFAR-10 (Krizhevsky et al., 2012). For the transfer learning of CIFAR-10, we use customized SGD optimizer with initial learning rate of 0.001 and momentum of 0.9 for $R_\theta$. MLP architectures of the discriminator network $D_\phi$ for MNIST, FashionMNIST and CIFAR-10 are given in Table A3. The 20-layer DenseNet networks shown in Table A9 were utlized for $R_\theta$ on the MNIST dataset, while the 100-layer DenseNet networks shown in Table A10 and A11 are fitted for $R_\theta$ on FashionMNIST and CIFAR-10.

Table A7: Hyper-parameters for the classification benchmark datasets

| Dataset | $\lambda$ | $d$ | $n$ | $T_1$ | $T_2$ | $s$ |
|---|---|---|---|---|---|---|
| MNIST | 1.0 | 16, 32, 64 | 64 | 1 | 300 | 0.1 |
| FashionMNIST | 1.0 | 16, 32, 64 | 64 | 1 | 300 | 1.0 |
| CIFAR-10 | 1.0 | 16, 32, 64 | 64 | 1 | 300 | 1.0 |
| CIFAR-10 (transfer learning) | 0.01 | 16, 32, 64 | 64 | 1 | 50 | 1.0 |

## B APPENDIX: PROOFS

In this appendix, we prove Lemmas 2.1 and 4.1, and Theorems 4.2 and 4.3.

### B.1 PROOF OF LEMMA 2.1

*Proof.* By assumption $\mu$ and $\gamma_d$ are both absolutely continuous with respect to the Lebesgue measure. The desired result holds since it is a spacial case of the well known results on the existence of optimal transport (Brenier, 1991; McCann, 1995), see, Theorem 1.28 on page 24 of (Philippis, 2013) for details. □

Table A8: Learning rate $\rho$ varies during training.

| Epoch | 0-150 | 151-225 | 226-300 |
|---|---|---|---|
| $\rho$ | 0.1 | 0.01 | 0.001 |

Table A9: Architecture for MNIST, reduced feature size is $d$

| Layers | Details | Output size |
|---|---|---|
| Convolution | $3 \times 3$ Conv | $24 \times 28 \times 28$ |
| Dense Block 1 | $\begin{bmatrix} \text{BN, } 1 \times 1 \text{ Conv} \\ \text{BN, } 3 \times 3 \text{ Conv} \end{bmatrix} \times 2$ | $48 \times 28 \times 28$ |
| Transition Layer 1 | BN, ReLU, $2 \times 2$ Average Pool,$1 \times 1$ Conv | $24 \times 14 \times 14$ |
| Dense Block 2 | $\begin{bmatrix} \text{BN, } 1 \times 1 \text{ Conv} \\ \text{BN, } 3 \times 3 \text{ Conv} \end{bmatrix} \times 2$ | $48 \times 14 \times 14$ |
| Transition Layer 2 | BN, ReLU, $2 \times 2$ Average Pool, $1 \times 1$ Conv | $24 \times 7 \times 7$ |
| Dense Block 3 | $\begin{bmatrix} \text{BN, } 1 \times 1 \text{ Conv} \\ \text{BN, } 3 \times 3 \text{ Conv} \end{bmatrix} \times 2$ | $48 \times 7 \times 7$ |
| Pooling | BN, ReLU, $7 \times 7$ Average Pool, Reshape | 48 |
| Fully connected | Linear | $d$ |

Table A10: Architecture for FashionMNIST, reduced feature size is $d$

| Layers | Details | Output size |
|---|---|---|
| Convolution | $3 \times 3$ Conv | $24 \times 28 \times 28$ |
| Dense Block 1 | $\begin{bmatrix} \text{BN, } 1 \times 1 \text{ Conv} \\ \text{BN, } 3 \times 3 \text{ Conv} \end{bmatrix} \times 16$ | $216 \times 28 \times 28$ |
| Transition Layer 1 | BN, ReLU, $2 \times 2$ Average Pool,$1 \times 1$ Conv | $108 \times 14 \times 14$ |
| Dense Block 2 | $\begin{bmatrix} \text{BN, } 1 \times 1 \text{ Conv} \\ \text{BN, } 3 \times 3 \text{ Conv} \end{bmatrix} \times 16$ | $300 \times 14 \times 14$ |
| Transition Layer 2 | BN, ReLU, $2 \times 2$ Average Pool, $1 \times 1$ Conv | $150 \times 7 \times 7$ |
| Dense Block 3 | $\begin{bmatrix} \text{BN, } 1 \times 1 \text{ Conv} \\ \text{BN, } 3 \times 3 \text{ Conv} \end{bmatrix} \times 16$ | $342 \times 7 \times 7$ |
| Pooling | BN, ReLU, $7 \times 7$ Average Pool, Reshape | 342 |
| Fully connected | Linear | $d$ |

Table A11: Architecture for CIFAR-10, reduced feature size is $d$

| Layers | Details | Output size |
|---|---|---|
| Convolution | $3 \times 3$ Conv | $24 \times 32 \times 32$ |
| Dense Block 1 | $\begin{bmatrix} \text{BN, } 1 \times 1 \text{ Conv} \\ \text{BN, } 3 \times 3 \text{ Conv} \end{bmatrix} \times 16$ | $216 \times 32 \times 32$ |
| Transition Layer 1 | BN, ReLU, $2 \times 2$ Average Pool,$1 \times 1$ Conv | $108 \times 16 \times 16$ |
| Dense Block 2 | $\begin{bmatrix} \text{BN, } 1 \times 1 \text{ Conv} \\ \text{BN, } 3 \times 3 \text{ Conv} \end{bmatrix} \times 16$ | $300 \times 16 \times 16$ |
| Transition Layer 2 | BN, ReLU, $2 \times 2$ Average Pool, $1 \times 1$ Conv | $150 \times 8 \times 8$ |
| Dense Block 3 | $\begin{bmatrix} \text{BN, } 1 \times 1 \text{ Conv} \\ \text{BN, } 3 \times 3 \text{ Conv} \end{bmatrix} \times 16$ | $342 \times 8 \times 8$ |
| Pooling | BN, ReLU, $8 \times 8$ Average Pool, Reshape | 342 |
| Fully connected | Linear | $d$ |

### B.2    PROOF OF LEMMA 4.1

*Proof.* Our proof follows Keziou (2003). Since $f(t)$ is convex, then $\forall t \in \mathbb{R}$, we have $f(t) = f^{**}(t)$, where

$$f^{**}(t) = \sup_{s \in \mathbb{R}} \{st - f^*(s)\}$$

is the Fenchel conjugate of $f^*$. By Fermat's rule, the maximizer $s^*$ satisfies

$$t \in \partial f^*(s^*),$$

i.e.,

$$s^* \in \partial f(t)$$

Plugging the above display with $t = \frac{d\mu_Z}{d\gamma}(x)$ into the definition of $f$-divergence, we derive (6).    □

### B.3    PROOF OF THEOREM 4.2

*Proof.* Without loss of generality, we assume $d = 1$. For $R^*$ satisfying (3) and any $R \in \mathcal{R}$, we have $R = \rho_{(R,R^*)}R^* + \varepsilon_R$, where $\rho_{(R,R^*)}$ is the correlation coefficient between $R$ and $R^*$, $\varepsilon_R = R - \rho_{(R,R^*)}R^*$. It is easy to see that $\varepsilon_R \perp\!\!\!\perp R^*$ and thus $Y \perp\!\!\!\perp \varepsilon_R$. As $(\rho_{(R,R^*)}R^*, Y)$ is independent of $(\varepsilon_R, 0)$, then by Theorem 3 of Székely & Rizzo (2009)

$$\begin{aligned}
\mathcal{V}[R, \mathbf{y}] &= \mathcal{V}[\rho_{(R,R^*)}R^* + \varepsilon_R, \mathbf{y}] \leq \mathcal{V}[\rho_{(R,R^*)}R^*, \mathbf{y}] + \mathcal{V}(\varepsilon_R, 0) \\
&= \mathcal{V}[\rho_{(R,R^*)}R^*, \mathbf{y}] = |\rho_{(R,R^*)}|\mathcal{V}[R^*, \mathbf{y}] \\
&\leq \mathcal{V}[R^*, \mathbf{y}].
\end{aligned}$$

As $R(\mathbf{x}) \sim \mathcal{N}(0,1)$ and $R^*(\mathbf{x}) \sim \mathcal{N}(0,1)$, then $\mathbb{D}_f(\mu_{R(\mathbf{x})}\|\gamma_d) = \mathbb{D}_f(\mu_{R^*(\mathbf{x})}\|\gamma_d) = 0$, and

$$\mathcal{L}(R) - \mathcal{L}(R^*) = \mathcal{V}[R^*, \mathbf{y}] - \mathcal{V}[R, \mathbf{y}] \geq 0.$$

The proof is completed.    □

### B.4    PROOF OF THEOREM 4.3

Denote $B_2 = \max\{|f'(c_1)|, |f'(c_2)|\}$, $B_3 = \max_{|s| \leq 2L_2\sqrt{d}\log n + B_2} |f^*(s)|$. We set the network parameters of the representer $R_\theta$ and the discriminator $D_\phi$ as follows.

(i) Representer network $\mathcal{R}_{\mathcal{D},\mathcal{W},\mathcal{S},\mathcal{B}}$ parameters: depth $\mathcal{D} = 9\log n + 12$, width $\mathcal{W} = d\max\{8d(n^{\frac{p}{2+p}}/\log n)^{\frac{1}{p}} + 4p, 12n^{\frac{p}{2+p}}/\log n + 14\}$, size $\mathcal{S} = dn^{\frac{p-2}{p+2}}/\log^4(npd)$, $\mathcal{B} = (2B_3L_1\sqrt{p} + \log n)\sqrt{d}$,

(ii) Discriminator network $\mathcal{M}_{\tilde{\mathcal{D}},\tilde{\mathcal{W}},\tilde{\mathcal{S}},\tilde{\mathcal{B}}}$ parameters: depth $\tilde{\mathcal{D}} = 9\log n + 12$, width $\tilde{\mathcal{W}} = \max\{8d(n^{\frac{d}{2+d}}/\log n)^{\frac{1}{d}} + 4d, 12n^{\frac{d}{2+d}}/\log n + 14\}$, size $\tilde{\mathcal{S}} = n^{\frac{d-2}{d+2}}/(\log^4 npd)$, $\tilde{\mathcal{B}} = 2L_2\sqrt{d}\log n + B_2$.

Before getting into the details of the proof of Theorem 4.3, we first give an outline of the basic structure of the proof.

Without loss of generality, we assume that $\lambda = 1$ and $m = 1$, i.e. $\mathbf{y} \in \mathbb{R}$. First we consider the scenario that $\mathbf{y}$ is bounded almost surely, say $|\mathbf{y}| \leq C_1$. We also assume $B_1 < \infty$. We can utilize the truncation technique to transfer the unbounded cases to the bounded ones under some common tail assumptions. Consequently, an additional $\log n$ multiplicative term will appear in the final results. For any $\bar{R} \in \mathcal{N}_{\mathcal{D},\mathcal{W},\mathcal{S},\mathcal{B}}$, we have,

$$\mathcal{L}(\widehat{R}_\theta) - \mathcal{L}(R^*) = \mathcal{L}(\widehat{R}_\theta) - \widehat{\mathcal{L}}(\widehat{R}_\theta) + \widehat{\mathcal{L}}(\widehat{R}_\theta) - \widehat{\mathcal{L}}(\bar{R}) + \widehat{\mathcal{L}}(\bar{R}) - \mathcal{L}(\bar{R}) + \mathcal{L}(\bar{R}) - \mathcal{L}(R^*)$$

$$\leq 2\sup_{R \in \mathcal{N}_{\mathcal{D},\mathcal{W},\mathcal{S},\mathcal{B}}} |\mathcal{L}(R) - \widehat{\mathcal{L}}(R)| + \inf_{\bar{R} \in \mathcal{N}_{\mathcal{D},\mathcal{W},\mathcal{S},\mathcal{B}}} |\mathcal{L}(\bar{R}) - \mathcal{L}(R^*)|, \qquad (10)$$

where we use the definition of $\widehat{R}_\theta$ in (8) and the feasibility of $\bar{R}$. Next we bound the two error terms in (10), i.e., **the approximation error** $\inf_{\bar{R} \in \mathcal{N}_{\mathcal{D},\mathcal{W},\mathcal{S},\mathcal{B}}} |\mathcal{L}(\bar{R}) - \mathcal{L}(R^*)|$ and **the statistical error** $\sup_{R \in \mathcal{N}_{\mathcal{D},\mathcal{W},\mathcal{S},\mathcal{B}}} |\mathcal{L}(R) - \widehat{\mathcal{L}}(R)|$ separately. Then Theorem 4.3 follows after bounding these two error terms.

### B.4.1 THE APPROXIMATION ERROR

**Lemma B.1.**
$$\inf_{\bar{R}\in\mathcal{N}_{\mathcal{D},\mathcal{W},\mathcal{S},\mathcal{B}}} |\mathcal{L}(\bar{R}) - \mathcal{L}(R^*)| \leq 2600 C_1 B_1 L_1 \sqrt{pd} n^{-\frac{2}{p+2}}. \tag{11}$$

*Proof.* By (3) and (6) and the definition of $\mathcal{L}$, we have

$$\inf_{\bar{R}\in\mathcal{N}_{\mathcal{D},\mathcal{W},\mathcal{S},\mathcal{B}}} |\mathcal{L}(\bar{R}) - \mathcal{L}(R^*)| \leq |\mathbb{D}_f(\mu_{\bar{R}_{\bar{\theta}}(\mathbf{x})}\|\gamma_d)| + |\mathcal{V}[R^*(\mathbf{x}),\mathbf{y}] - \mathcal{V}[\bar{R}_{\bar{\theta}}(\mathbf{x}),\mathbf{y}]|, \tag{12}$$

where $\bar{R}_{\bar{\theta}} \in \mathcal{N}_{\mathcal{D},\mathcal{W},\mathcal{S},\mathcal{B}}$ is specified in Lemma B.2 below. We finish the proof by (14) in Lemma B.3 and (15) in Lemma B.4, which will be proved below. $\square$

**Lemma B.2.** *Define* $\tilde{R}^*(x) = \min\{R^*(x), \log n\}$. *There exist a* $\bar{R}_{\bar{\theta}} \in \mathcal{N}_{\mathcal{D},\mathcal{W},\mathcal{S},\mathcal{B}}$ *with depth* $\mathcal{D} = 9\log n + 12$, *width* $\mathcal{W} = d\max\{8d(n^{\frac{p}{2+p}}/\log n)^{\frac{1}{p}} + 4d, 12n^{\frac{p}{2+p}}/\log n + 14\}$, *and size* $\mathcal{S} = dn^{\frac{p-2}{p+2}}/(\log^4 npd)$, $\mathcal{B} = (2B_1 L_1\sqrt{p} + \log n)\sqrt{d}$, *such that*

$$\|\bar{R}_{\bar{\theta}} - \tilde{R}^*\|_{L^2(\mu_{\mathbf{x}})} \leq 160 L_1 B_1 \sqrt{pd} n^{-\frac{2}{p+2}}. \tag{13}$$

*Proof.* Let $\tilde{R}_i^*(x)$ be the $i$-th entry of $\tilde{R}^*(x) : \mathbb{R}^d \to \mathbb{R}^d$. By the assumption on $R^*$, it is easy to check that $\tilde{R}_i^*(x)$ is Lipschitz continuous on $[-B_1, B_1]^d$ with the Lipschitz constant $L_1$ and $\|\tilde{R}_i^*\|_{L^\infty} \leq \log n$. By Theorem 4.3 in Shen et al. (2019), there exists a ReLU network $\bar{R}_{\bar{\theta}_i}$ with with depth $9\log n + 12$, width $\max\{8d(n^{\frac{p}{2+p}}/\log n)^{\frac{1}{p}} + 4d, 12n^{\frac{p}{2+p}}/\log n + 14\}$, $\|\bar{R}_{\bar{\theta}_i}\|_{L^\infty} = 2B_1 L_1\sqrt{p} + \log n$, such that

$$\|\bar{R}_{\bar{\theta}_i}\|_{L^\infty} \leq 2B_1 L_1\sqrt{p} + \log n,$$

and

$$\|\tilde{R}_i^* - \bar{R}_{\bar{\theta}_i}\|_{L^\infty([-B_1,B_1]^p\backslash\mathcal{H})} \leq 80 L_1 B_1\sqrt{p} n^{-\frac{2}{p+2}},$$

$$\mu_{\mathbf{x}}(\mathcal{H}) \leq \frac{80 L_1 B_1\sqrt{p} n^{-\frac{2}{p+2}}}{2B_1 L_1\sqrt{p} + \log n}.$$

Define $\bar{R}_{\bar{\theta}} = [\bar{R}_{\bar{\theta}_1}, \ldots, \bar{R}_{\bar{\theta}_d}] \in \mathcal{N}_{\mathcal{D},\mathcal{W},\mathcal{S},\mathcal{B}}$. The above three display implies

$$\|\bar{R}_{\bar{\theta}} - \tilde{R}^*\|_{L^2(\mu_{\mathbf{x}})} \leq 160 L_1 B_1\sqrt{pd} n^{-\frac{2}{p+2}}.$$

$\square$

**Lemma B.3.**
$$|\mathcal{V}[R^*(\mathbf{x}),\mathbf{y}] - \mathcal{V}[\bar{R}_{\bar{\theta}}(\mathbf{x}),\mathbf{y}]| \leq 2580 C_1 B_1 L_1 \sqrt{pd} n^{-\frac{2}{p+2}}. \tag{14}$$

*Proof.* Recall that Székely et al. (2007)

$$\mathcal{V}[\mathbf{z},\mathbf{y}] = \mathbb{E}\left[\|\mathbf{z}_1 - \mathbf{z}_2\|\|\mathbf{y}_1 - \mathbf{y}_2\|\right] - 2\mathbb{E}\left[\|\mathbf{z}_1 - \mathbf{z}_2\|\|\mathbf{y}_1 - \mathbf{y}_3\|\right]$$
$$+ \mathbb{E}\left[\|\mathbf{z}_1 - \mathbf{z}_2\|\right]\mathbb{E}\left[\|\mathbf{y}_1 - \mathbf{y}_2\|\right],$$

where $(\mathbf{z}_i, \mathbf{y}_i), i = 1, 2, 3$ are i.i.d. copies of $(\mathbf{z}, \mathbf{y})$. We have

$$|\mathcal{V}[R^*(\mathbf{x}),\mathbf{y}] - \mathcal{V}[\bar{R}_{\bar{\theta}}(\mathbf{x}),\mathbf{y}]|$$
$$\leq |\mathbb{E}\left[(\|R^*(\mathbf{x}_1) - R^*(\mathbf{x}_2)\| - \|\bar{R}_{\bar{\theta}}(\mathbf{x}_1) - \bar{R}_{\bar{\theta}}(\mathbf{x}_2)\|)|\mathbf{y}_1 - \mathbf{y}_2|\right]|$$
$$+ 2|\mathbb{E}\left[(\|R^*(\mathbf{x}_1) - R^*(\mathbf{x}_2)\| - \|\bar{R}_{\bar{\theta}}(\mathbf{x}_1) - \bar{R}_{\bar{\theta}}(\mathbf{x}_2)\|)|\mathbf{y}_1 - \mathbf{y}_3|\right]|$$
$$+ |\mathbb{E}\left[\|R^*(\mathbf{x}_1) - R^*(\mathbf{x}_2)\| - \|\bar{R}_{\bar{\theta}}(\mathbf{x}_1) - \bar{R}_{\bar{\theta}}(\mathbf{x}_2)\|\right]\mathbb{E}\left[\|\mathbf{y}_1 - \mathbf{y}_2\|\right]|$$
$$\leq 8C_1\mathbb{E}\left[|\|R^*(\mathbf{x}_1) - R^*(\mathbf{x}_2)\| - \|\bar{R}_{\bar{\theta}}(\mathbf{x}_1) - \bar{R}_{\bar{\theta}}(\mathbf{x}_2)\||\right]$$
$$\leq 16C_1\mathbb{E}\left[|\|R^*(\mathbf{x}) - \bar{R}_{\bar{\theta}}(\mathbf{x})\||\right]$$
$$\leq 16C_1(\mathbb{E}\left[\|\tilde{R}^*(\mathbf{x}) - \bar{R}_{\bar{\theta}}(\mathbf{x})\|\right] + \mathbb{E}\left[\|R^*(\mathbf{x})\mathbf{1}_{R^*(\mathbf{x})\in\text{Ball}^c(\mathbf{0},\log n)}\|\right]),$$

where in the first and third inequalities we use the triangle inequality, and second one follows from the boundedness of $\mathbf{y}$. By (13), the first term in the last line is bounded by $2560C_1B_1L_1\sqrt{pd}n^{-\frac{1}{p+2}}$. Some direct calculation shows that

$$\mathbb{E}\left[\|R^*(\mathbf{x})\mathbf{1}_{R^*(\mathbf{x})\in\text{Ball}^c(\mathbf{0},\log n)}\|\right] \le C_2\frac{(\log n)^d}{n}.$$

We finish the proof by comparing the order of the above two terms, i.e., $C_2\frac{(\log n)^d}{n} \le 20C_1B_1L_1\sqrt{pd}n^{-\frac{2}{p+2}}$. $\qquad\square$

**Lemma B.4.**

$$|\mathbb{D}_f(\mu_{\bar{R}_{\bar\theta}(\mathbf{x})}\|\gamma_d)| \le 20C_1B_1L_1\sqrt{pd}n^{-\frac{2}{p+2}}. \tag{15}$$

*Proof.* By Lemma B.2 $\bar{R}_{\bar\theta}$ can approximate $R^*$ arbitrary well, the desired result follows from the fact that $\mathbb{D}_f(\mu_{R^*(\mathbf{x})}\|\gamma_d) = 0$ and the continuity of $\mathbb{D}_f(\mu_{R(\mathbf{x})}\|\gamma_d)$ on $R$. We present the sketch of the proof and omit the details here. Let $r^*(z) = \frac{\mathrm{d}\mu_{R^*(\mathbf{x})}}{\mathrm{d}\gamma_d}(z)$ and $\bar{r}(z) = \frac{\mathrm{d}\mu_{\bar{R}_{\bar\theta}(\mathbf{x})}}{\mathrm{d}\gamma_d}(z)$. By definition we have

$$\begin{aligned}
\mathbb{D}_f(\mu_{R^*(\mathbf{x})}\|\gamma_d) &= \mathbb{E}_{W\sim\gamma^d}[f(r^*(W))]\\
&= \mathbb{E}_{W\sim\gamma^d}[f(r^*(W))\mathbf{1}_{W\in\text{Ball}(0,\log n)}] + \mathbb{E}_{W\sim\gamma^d}[f(r^*(W))\mathbf{1}_{W\in\text{Ball}^c(0,\log n)}].
\end{aligned}$$

(We can represent $\mathbb{D}_f(\mu_{\bar{R}_{\bar\theta}}\|\gamma_d)$ similarly. ) Then

$$\begin{aligned}
&|\mathbb{D}_f(\mu_{\bar{R}_{\bar\theta}(\mathbf{x})}\|\gamma_d)| = |\mathbb{D}_f(\mu_{\bar{R}_{\bar\theta}(\mathbf{x})}\|\gamma_d) - \mathbb{D}_f(\mu_{R^*(\mathbf{x})}\|\gamma_d)|\\
&\le \mathbb{E}_{W\sim\gamma^d}[|f(r^*(W)) - f(\bar{r}(W))|\mathbf{1}_{W\in\text{Ball}(0,\log n)}]\\
&+ \mathbb{E}_{W\sim\gamma^d}[|f(r^*(W)) - f(r^*(W))|\mathbf{1}_{W\in\text{Ball}^c(0,\log n)}]\\
&\le \int_{\|z\|\le\log n}|f'(\tilde{r}(z))||r^*(z) - \bar{r}(z)|\mathrm{d}\gamma_d(z) + \int_{\|z\|>\log n}|f'(\tilde{r}(z))||r^*(z) - \bar{r}(z)|\mathrm{d}\gamma_d(z)\\
&\le C_3\int_{\|z\|\le\log n}|r^*(z) - \bar{r}(z)|\mathrm{d}\gamma_d(z) + C_4\int_{\|z\|>\log n}|r^*(z) - \bar{r}(z)|
\end{aligned}$$

The first term in the above display is small due to $\bar{R}_{\bar\theta}$ can approximate $R^*$ well. The second term is small due to the boundedness of $\bar{r}$ and the exponential decay of the Gaussian tails. $\qquad\square$

### B.4.2 The statistical error

**Lemma B.5.**

$$\sup_{R\in\mathcal{N}_{\mathcal{D},\mathcal{W},\mathcal{S},\mathcal{B}}}|\mathcal{L}(R) - \widehat{\mathcal{L}}(R)| \le C_{15}(B_1(L_1 + L2)\sqrt{pd})n^{-\frac{2}{2+p}} + (L_2\sqrt{d} + B_2 + B_3)\log n\,n^{-\frac{2}{2+d}}) \tag{16}$$

*Proof.* By the definition and the triangle inequality we have

$$\begin{aligned}
&\mathbb{E}[\sup_{R\in\mathcal{N}_{\mathcal{D},\mathcal{W},\mathcal{S},\mathcal{B}}}|\mathcal{L}(R) - \widehat{\mathcal{L}}(R)|]\\
&\le \mathbb{E}[\sup_{R\in\mathcal{N}_{\mathcal{D},\mathcal{W},\mathcal{S},\mathcal{B}}}|\widehat{\mathcal{V}}_n[R(\mathbf{x}),\mathbf{y}] - \mathcal{V}[(R(\mathbf{x}),\mathbf{y})]|]\\
&+ \mathbb{E}[\sup_{R\in\mathcal{N}_{\mathcal{D},\mathcal{W},\mathcal{S},\mathcal{B}}}|\widehat{\mathbb{D}}_f(\mu_{R(\mathbf{x})}\|\gamma_d) - \mathbb{D}_f(\mu_{R(\mathbf{x})}\|\gamma_d)|].
\end{aligned}$$

We finish the proof based on (17) in Lemma B.6 and (22) in Lemma B.7, which will be proved below. $\qquad\square$

**Lemma B.6.**

$$\mathbb{E}[\sup_{R\in\mathcal{N}_{\mathcal{D},\mathcal{W},\mathcal{S},\mathcal{B}}}|\widehat{\mathcal{V}}_n[R(\mathbf{x}),\mathbf{y}] - \mathcal{V}[R(\mathbf{x}),\mathbf{y}]|] \le 4C_6C_7C_{10}B_1L_1\sqrt{pd}n^{-\frac{2}{p+2}}. \tag{17}$$

*Proof.* We first fix some notation for simplicity. Denote $O = (\mathbf{x}, \mathbf{y}) \in \mathbb{R}^p \times \mathbb{R}^1$ and $O_i = (\mathbf{x}_i, \mathbf{y}_i), i = 1, \ldots n$ are i.i.d copy of $O$, and denote $\mu_{\mathbf{x}, \mathbf{y}}$ and $\mathbb{P}^{\otimes n}$ as $\mathbb{P}$ and $\mathbb{P}^n$, respectively. $\forall R \in \mathcal{N}_{\mathcal{D}, \mathcal{W}, \mathcal{S}, \mathcal{B}}$, let $\tilde{O} = (R(\mathbf{x}), \mathbf{y})$ and $\tilde{O}_i = (R(\mathbf{x}_i), \mathbf{y}_i), i = 1, \ldots n$ are i.i.d copy of $\tilde{O}$. Define centered kernel $\bar{h}_R : (\mathbb{R}^p \times \mathbb{R}^1)^{\otimes 4} \to \mathbb{R}$ as

$$
\begin{aligned}
\bar{h}_R(\tilde{O}_1, \tilde{O}_2, \tilde{O}_3, \tilde{O}_4) &= \tfrac{1}{4} \sum_{\substack{1 \leq i,j \leq 4, \\ i \neq j}} \|R(\mathbf{x}_i) - R(\mathbf{x}_j)\| |\mathbf{y}_i - \mathbf{y}_j| \\
&- \tfrac{1}{4} \sum_{i=1}^4 \left( \sum_{\substack{1 \leq j \leq 4, \\ j \neq i}} \|R(\mathbf{x}_i) - R(\mathbf{x}_j)\| \sum_{\substack{1 \leq j \leq 4, \\ i \neq j}} |\mathbf{y}_i - \mathbf{y}_j| \right) \\
&+ \tfrac{1}{24} \sum_{\substack{1 \leq i,j \leq 4, \\ i \neq j}} \|R(\mathbf{x}_i) - R(\mathbf{x}_j)\| \sum_{\substack{1 \leq i,j \leq 4, \\ i \neq j}} |\mathbf{y}_i - \mathbf{y}_j| - \mathcal{V}[R(\mathbf{x}), \mathbf{y}]
\end{aligned} \tag{18}
$$

Then, the centered $U$-statistics $\widehat{\mathcal{V}}_n[R(\mathbf{x}), \mathbf{y}] - \mathcal{V}[R(\mathbf{x}), \mathbf{y}]$ can be represented as

$$
\mathbb{U}_n(\bar{h}_R) = \frac{1}{C_n^4} \sum_{1 \leq i_1 < i_2 < i_3 < i_4 \leq n} \bar{h}_R(\tilde{O}_{i_1}, \tilde{O}_{i_2}, \tilde{O}_{i_3}, \tilde{O}_{i_4}).
$$

Our goal is to bound the supremum of the centered $U$-process $\mathbb{U}_n(\bar{h}_R)$ with the nondegenerate kernel $\bar{h}_R$. By the symmetrization randomization Theorem 3.5.3 in De la Pena & Giné (2012), we have

$$
\mathbb{E}[\sup_{R \in \mathcal{N}_{\mathcal{D}, \mathcal{W}, \mathcal{S}, \mathcal{B}}} |\mathbb{U}_n(\bar{h}_R)|] \leq C_5 \mathbb{E}[\sup_{R \in \mathcal{N}_{\mathcal{D}, \mathcal{W}, \mathcal{S}, \mathcal{B}}} |\frac{1}{C_n^4} \sum_{1 \leq i_1 < i_2 < i_3 < i_4 \leq n} \epsilon_{i_1} \bar{h}_R(\tilde{O}_{i_1}, \tilde{O}_{i_2}, \tilde{O}_{i_3}, \tilde{O}_{i_4})|],
$$
(19)

where, $\epsilon_{i_1}, i_1 = 1, \ldots n$ are i.i.d Rademacher variables that are also independent with $\tilde{O}_i, i = 1, \ldots, n$. We finish the proof by upper bounding the above Rademacher process with the matric entropy of $\mathcal{N}_{\mathcal{D}, \mathcal{W}, \mathcal{S}, \mathcal{B}}$. To this end we need the following lemma.

**Lemma B.7.** *If $\xi_i, i = 1, \ldots m$ are $m$ finite linear combinations of Rademacher variables $\epsilon_j, j = 1, \ldots J$. Then*

$$
\mathbb{E}_{\epsilon_j, j=1, \ldots J} \max_{1 \leq i \leq m} |\xi_i| \leq C_6 (\log m)^{1/2} \max_{1 \leq i \leq m} \left( \mathbb{E} \xi_i^2 \right)^{1/2}. \tag{20}
$$

*Proof.* This result follows directly from Corollary 3.2.6 and inequality (4.3.1) in De la Pena & Giné (2012) with $\Phi(x) = \exp(x^2)$. □

By the boundedness assumption on $\mathbf{y}$ and the boundedness of $R \in \mathcal{N}_{\mathcal{D}, \mathcal{W}, \mathcal{S}, \mathcal{B}}$, we have that the kernel $\bar{h}_R$ is also bounded, say

$$
\|\bar{h}_R\|_{L^\infty} \leq C_7 (2B_1 L_1 \sqrt{p} + \log n) \sqrt{d}. \tag{21}
$$

$\forall R, \tilde{R} \in \mathcal{N}_{\mathcal{D}, \mathcal{W}, \mathcal{S}, \mathcal{B}}$ define a random empirical measure (depends on $O_i, i = 1, \ldots, n$)

$$
e_{n,1}(R, \tilde{R}) = \mathbb{E}_{\epsilon_{i_1}, i_1=1, \ldots, n} |\frac{1}{C_n^4} \sum_{1 \leq i_1 < i_2 < i_3 < i_4 \leq n} \epsilon_{i_1} (\bar{h}_R - \bar{h}_{\tilde{R}})(\tilde{O}_{i_1}, \ldots, \tilde{O}_{i_4})|.
$$

Condition on $O_i, i = 1, \ldots, n$, let $\mathfrak{C}(\mathcal{N}, e_{n,1}, \delta))$ be the covering number of $\mathcal{N}_{\mathcal{D}, \mathcal{W}, \mathcal{S}, \mathcal{B}}$ with respect to the empirical distance $e_{n,1}$ at scale of $\delta > 0$. Denote $\mathcal{N}_\delta$ as the covering set of $\mathcal{N}_{\mathcal{D}, \mathcal{W}, \mathcal{S}, \mathcal{B}}$ with

cardinality of $\mathfrak{C}(\mathcal{N}, e_{n,1}, \delta))$. Then,

$$\mathbb{E}_{\epsilon_{i_1}}\big[\sup_{R\in\mathcal{N}_{\mathcal{D},\mathcal{W},\mathcal{S},\mathcal{B}}} |\frac{1}{C_n^4}\sum_{1\le i_1<i_2<i_3<i_4\le n}\epsilon_{i_1}\bar{h}_R(\tilde{O}_{i_1},\tilde{O}_{i_2},\tilde{O}_{i_3},\tilde{O}_{i_4})|\big]$$

$$\le \delta + \mathbb{E}_{\epsilon_{i_1}}\big[\sup_{R\in\mathcal{N}_{\delta}} |\frac{1}{C_n^4}\sum_{1\le i_1<i_2<i_3<i_4\le n}\epsilon_{i_1}\bar{h}_R(\tilde{O}_{i_1},\tilde{O}_{i_2},\tilde{O}_{i_3},\tilde{O}_{i_4})|\big]$$

$$\le \delta + C_6\frac{1}{C_n^4}(\log\mathfrak{C}(\mathcal{N},e_{n,1},\delta))^{1/2}\max_{R\in\mathcal{N}_{\delta}}\big[\sum_{i_1=1}^{n}\sum_{i_2<i_3<i_4}(\bar{h}_R(\tilde{O}_{i_1},\tilde{O}_{i_2},\tilde{O}_{i_3},\tilde{O}_{i_4}))^2\big]^{1/2}$$

$$\le \delta + C_6C_7(2B_1L_1\sqrt{p}+\log n)\sqrt{d}(\log\mathfrak{C}(\mathcal{N},e_{n,1},\delta))^{1/2}\frac{1}{C_n^4}\big[\frac{n(n!)^2}{((n-3)!)^2}\big]^{1/2}$$

$$\le \delta + 2C_6C_7(2B_1L_1\sqrt{p}+\log n)\sqrt{d}(\log\mathfrak{C}(\mathcal{N},e_{n,1},\delta))^{1/2}/\sqrt{n}$$

$$\le \delta + 2C_6C_7(2B_1L_1\sqrt{p}+\log n)\sqrt{d}(\text{VC}_{\mathcal{N}}\log\frac{2e\mathcal{B}n}{\delta\text{VC}_{\mathcal{N}}})^{1/2}/\sqrt{n}$$

$$\le \delta + C_6C_7C_{10}(B_1L_1\sqrt{p}+\log n)\sqrt{d}(\mathcal{D}\mathcal{S}\log\mathcal{S}\log\frac{\mathcal{B}n}{\delta\mathcal{D}\mathcal{S}\log\mathcal{S}})^{1/2}/\sqrt{n}.$$

where the first inequality follows from the triangle inequality, the second inequality uses (20), the third and fourth inequalities follow after some algebra, and the fifth inequality holds due to $\mathfrak{C}(\mathcal{N}, e_{n,1}, \delta) \le \mathfrak{C}(\mathcal{N}, e_{n,\infty}, \delta)$ and the relationship between the metric entropy and the VC-dimension of the ReLU networks $\mathcal{N}_{\mathcal{D},\mathcal{W},\mathcal{S},\mathcal{B}}$ (Anthony & Bartlett, 2009), i.e.,

$$\log\mathfrak{C}(\mathcal{N}, e_{n,\infty}, \delta)) \le \text{VC}_{\mathcal{N}}\log\frac{2e\mathcal{B}n}{\delta\text{VC}_{\mathcal{N}}},$$

and the last inequality holds due to the upper bound of VC-dimension for the ReLU network $\mathcal{N}_{\mathcal{D},\mathcal{W},\mathcal{S},\mathcal{B}}$ satisfying

$$C_8\mathcal{D}\mathcal{S}\log\mathcal{S} \le \text{VC}_{\mathcal{N}} \le C_9\mathcal{D}\mathcal{S}\log\mathcal{S},$$

see Bartlett et al. (2019). Then (17) holds by the selection of the network parameters and set $\delta = \frac{1}{n}$ and some algebra. $\qquad\square$

**Lemma B.8.**

$$\mathbb{E}\big[\sup_{R\in\mathcal{N}_{\mathcal{D},\mathcal{W},\mathcal{S},\mathcal{B}}} |\widehat{\mathbb{D}}_f(\mu_{R(\mathbf{x})}||\gamma_d) - \mathbb{D}_f(\mu_{R(\mathbf{x})}||\gamma_d)|\big] \le C_{14}(L_2\sqrt{d}+B_2+B_3)(n^{-\frac{2}{2+p}}+\log n\, n^{-\frac{2}{2+d}}) \quad (22)$$

*Proof.* $\forall R \in \mathcal{N}_{\mathcal{D},\mathcal{W},\mathcal{S},\mathcal{B}}$, let $r(z) = \frac{\mathrm{d}\mu_{R(\mathbf{x})}}{\mathrm{d}\gamma_d}(z)$, $g_R(z) = f'(r(z))$. By assumption $g_R(z) : \mathbb{R}^d \to \mathbb{R}$ is Lipschitz continuous with the Lipschitz constant $L_2$ and $\|g_R\|_{L^\infty} \le B_2$. Without loss of generality, we assume $\text{supp}(g_R) \subseteq [-\log n, \log n]^d$. Then, similar to the proof of Lemma B.2 we can show that there exists a $\bar{D}_{\bar{\phi}} \in \mathcal{M}_{\tilde{\mathcal{D}},\tilde{\mathcal{W}},\tilde{\mathcal{S}},\tilde{\mathcal{B}}}$ with depth $\tilde{\mathcal{D}} = 9\log n + 12$, width $\tilde{\mathcal{W}} = \max\{8d(n^{\frac{d}{2+d}}/\log n)^{\frac{1}{d}} + 4d, 12n^{\frac{d}{2+d}}/\log n + 14\}$, and size $\tilde{\mathcal{S}} = n^{\frac{d-2}{d+2}}/(\log^4 npd)$, $\tilde{\mathcal{B}} = 2L_2\sqrt{d}\log n + B_2$ such that for $\mathbf{z} \sim \gamma_d$ and $\mathbf{z} \sim \mu_{R(\mathbf{x})}$

$$\mathbb{E}_{\mathbf{z}}[|\bar{D}_{\bar{\phi}}(\mathbf{z}) - g_R(\mathbf{z})|] \le 160L_2\sqrt{d}\log n\, n^{-\frac{2}{d+2}}. \quad (23)$$

$\forall g : \mathbb{R}^d \to \mathbb{R}$, define

$$\mathcal{E}(g) = \mathbb{E}_{\mathbf{x}\sim\mu_{\mathbf{x}}}[g(R(\mathbf{x}))] - \mathbb{E}_{W\sim\gamma_d}[f^*(g(W))],$$

$$\widehat{\mathcal{E}}(g) = \widehat{\mathcal{E}}(g, R) = \frac{1}{n}\sum_{i=1}^{n}[g(R(\mathbf{x}_i)) - f^*(g(W_i))].$$

By (6) we have

$$\mathcal{E}(g_R) = \mathbb{D}_f(\mu_{R(\mathbf{x})}||\gamma_d) = \sup_{\text{measureable } D:\mathbb{R}^d\to\mathbb{R}} \mathcal{E}(D). \quad (24)$$

Then,

$$|\mathbb{D}_f(\mu_{R(\mathbf{x})}||\gamma_d) - \widehat{\mathbb{D}}_f(\mu_{R(\mathbf{x})}||\gamma_d)|$$

$$= |\mathcal{E}(g_R) - \max_{D_\phi \in \mathcal{M}_{\tilde{\mathcal{D}},\tilde{\mathcal{W}},\tilde{\mathcal{S}},\tilde{\mathcal{B}}}} \widehat{\mathcal{E}}(D_\phi)|$$

$$\leq |\mathcal{E}(g_R) - \sup_{D_\phi \in \mathcal{M}_{\tilde{\mathcal{D}},\tilde{\mathcal{W}},\tilde{\mathcal{S}},\tilde{\mathcal{B}}}} \mathcal{E}(D_\phi)| + |\sup_{D_\phi \in \mathcal{M}_{\tilde{\mathcal{D}},\tilde{\mathcal{W}},\tilde{\mathcal{S}},\tilde{\mathcal{B}}}} \mathcal{E}(D_\phi) - \max_{D_\phi \in \mathcal{M}_{\tilde{\mathcal{D}},\tilde{\mathcal{W}},\tilde{\mathcal{S}},\tilde{\mathcal{B}}}} \widehat{\mathcal{E}}(D_\phi)|$$

$$\leq |\mathcal{E}(g_R) - \mathcal{E}(\bar{D}_{\bar{\phi}})| + \sup_{D_\phi \in \mathcal{M}_{\tilde{\mathcal{D}},\tilde{\mathcal{W}},\tilde{\mathcal{S}},\tilde{\mathcal{B}}}} |\mathcal{E}(D_\phi) - \widehat{\mathcal{E}}(D_\phi)|$$

$$\leq \mathbb{E}_{\mathbf{z} \sim \mu_{R(\mathbf{x})}}[|g_R - \bar{D}_{\bar{\phi}}|(\mathbf{z})] + \mathbb{E}_{W \sim \gamma_d}[|f^*(g_R) - f^*(\bar{D}_{\bar{\phi}})|(W)] + \sup_{D_\phi \in \mathcal{M}_{\tilde{\mathcal{D}},\tilde{\mathcal{W}},\tilde{\mathcal{S}},\tilde{\mathcal{B}}}} |\mathcal{E}(D_\phi) - \widehat{\mathcal{E}}(D_\phi)|$$

$$\leq 160(1 + B_3)L_2\sqrt{d} \log n \, n^{-\frac{2}{d+2}} + \sup_{D_\phi \in \mathcal{M}_{\tilde{\mathcal{D}},\tilde{\mathcal{W}},\tilde{\mathcal{S}},\tilde{\mathcal{B}}}} |\mathcal{E}(D_\phi) - \widehat{\mathcal{E}}(D_\phi)|$$

where we use the triangle inequality in the first inequality, and we use $\mathcal{E}(g_R) \geq \sup_{D_\phi \in \mathcal{M}_{\tilde{\mathcal{D}},\tilde{\mathcal{W}},\tilde{\mathcal{S}},\tilde{\mathcal{B}}}} \mathcal{E}(D_\phi)$ followed from (24) and the triangle inequality in the second inequality, the third inequality follows from the triangle inequality, and the last inequality follows from (23) and the mean value theorem. We finish the proof via bounding the empirical process

$$\mathbb{U}(D, R) = \mathbb{E}[\sup_{R \in \mathcal{N}_{\mathcal{D},\mathcal{W},\mathcal{S},\mathcal{B}}, D \in \mathcal{M}_{\tilde{\mathcal{D}},\tilde{\mathcal{W}},\tilde{\mathcal{S}},\tilde{\mathcal{B}}}} |\mathcal{E}(D) - \widehat{\mathcal{E}}(D)|].$$

Let $S = (\mathbf{x}, \mathbf{z}) \sim \mu_{\mathbf{x}} \bigotimes \gamma_d$ and $S_i, i = 1, \ldots, n$ be $n$ i.i.d copy of $S$. Denote

$$b(D, R; S) = D(R(\mathbf{x})) - f^*(D(\mathbf{z})).$$

Then

$$\mathcal{E}(D, R) = \mathbb{E}_S[b(D, R; S)]$$

and

$$\widehat{\mathcal{E}}(D, R) = \frac{1}{n} \sum_{i=1}^n b(D, R; S_i).$$

Let

$$\mathcal{G}(\mathcal{M} \times \mathcal{N}) = \frac{1}{n} \mathbb{E}_{\{S_i, \epsilon_i\}_i^n} \left[ \sup_{R \in \mathcal{N}_{\mathcal{D},\mathcal{W},\mathcal{S},\mathcal{B}}, D \in \mathcal{M}_{\tilde{\mathcal{D}},\tilde{\mathcal{W}},\tilde{\mathcal{S}},\tilde{\mathcal{B}}}} |\sum_{i=1}^n \epsilon_i b(D, R; S_i)| \right]$$

be the Rademacher complexity of $\mathcal{M}_{\tilde{\mathcal{D}},\tilde{\mathcal{W}},\tilde{\mathcal{S}},\tilde{\mathcal{B}}} \times \mathcal{N}_{\mathcal{D},\mathcal{W},\mathcal{S},\mathcal{B}}$ (Bartlett & Mendelson, 2002). Let $\mathfrak{C}(\mathcal{M} \times \mathcal{N}, e_{n,1}, \delta))$ be the covering number of $\mathcal{M}_{\tilde{\mathcal{D}},\tilde{\mathcal{W}},\tilde{\mathcal{S}},\tilde{\mathcal{B}}} \times \mathcal{N}_{\mathcal{D},\mathcal{W},\mathcal{S},\mathcal{B}}$ with respect to the empirical distance (depends on $S_i$)

$$d_{n,1}((D, R), (\tilde{D}, \tilde{R})) = \frac{1}{n} \mathbb{E}_{\epsilon_i}[\sum_{i=1}^n |\epsilon_i(b(D, R; S_i) - b(\tilde{D}, \tilde{R}; S_i))|]$$

at scale of $\delta > 0$. Let $\mathcal{M}_\delta \times \mathcal{N}_\delta$ be such a converging set of $\mathcal{M}_{\tilde{\mathcal{D}},\tilde{\mathcal{W}},\tilde{\mathcal{S}},\tilde{\mathcal{B}}} \times \mathcal{N}_{\mathcal{D},\mathcal{W},\mathcal{S},\mathcal{B}}$. Then,

$$\mathbb{U}(D, R) = 2\mathcal{G}(\mathcal{M} \times \mathcal{N})$$

$$= 2\mathbb{E}_{S_1,\ldots,S_n}[\mathbb{E}_{\epsilon_i, i=1,\ldots,n}[\mathcal{G}(\mathcal{N} \times \mathcal{M})|(S_1, \ldots, S_n)]]$$

$$\leq 2\delta + \frac{2}{n} \mathbb{E}_{S_1,\ldots,S_n}[\mathbb{E}_{\epsilon_i, i=1,\ldots,n}[\sup_{(D,R) \in \mathcal{M}_\delta \times \mathcal{N}_\delta} |\sum_{i=1}^n \epsilon_i b(D, R; S_i)||(S_1, \ldots, S_n)]]$$

$$\leq 2\delta + C_{12} \frac{1}{n} \mathbb{E}_{S_1,\ldots,S_n}[(\log \mathfrak{C}(\mathcal{M} \times \mathcal{N}, e_{n,1}, \delta))^{1/2} \max_{(D,R) \in \mathcal{M}_\delta \times \mathcal{N}_\delta} [\sum_{i=1}^n b^2(D, R; S_i)]^{1/2}]$$

$$\leq 2\delta + C_{12} \frac{1}{n} \mathbb{E}_{S_1,\ldots,S_n}[(\log \mathfrak{C}(\mathcal{M} \times \mathcal{N}, e_{n,1}, \delta))^{1/2} \sqrt{n}(2L_2\sqrt{d} \log n + B_2 + B_3)]$$

$$\leq 2\delta + C_{12} \frac{1}{\sqrt{n}}(2L_2\sqrt{d} \log n + B_2 + B_3)(\log \mathfrak{C}(\mathcal{M}, e_{n,1}, \delta) + \log \mathfrak{C}(\mathcal{N}, d_{n,1}, \delta))^{1/2}$$

$$\leq 2\delta + C_{13} \frac{L_2\sqrt{d} \log n + B_2 + B_3}{\sqrt{n}}(\mathcal{D}\mathcal{S} \log \mathcal{S} \log \frac{\mathcal{B}n}{\delta \mathcal{D}\mathcal{S} \log \mathcal{S}} + \tilde{\mathcal{D}}\tilde{\mathcal{S}} \log \tilde{\mathcal{S}} \log \frac{\tilde{\mathcal{B}}n}{\delta \tilde{\mathcal{D}}\tilde{\mathcal{S}} \log \tilde{\mathcal{S}}})^{1/2}$$

where the first equality follows from the standard symmetrization technique, the second equality holds due to the iteration law of conditional expectation, the first inequality follows from the triangle inequality, and the second inequality uses equation 20, the third inequality uses the fact that $b(D, R; S)$ is bounded, i.e., $\|b(D, R; S)\|_{L^\infty} \leq 2L_2\sqrt{d}\log n + B_2 + B_3$, and the fourth inequality follows from some algebra, and the fifth inequality follows from $\mathfrak{C}(\mathcal{N}, e_{n,1}, \delta) \leq \mathfrak{C}(\mathcal{N}, e_{n,\infty}, \delta)$ (similar result for $\mathcal{M}$) and $\log \mathfrak{C}(\mathcal{N}, e_{n,\infty}, \delta)) \leq \mathrm{VC}_\mathcal{N} \log \frac{2e\mathcal{B}n}{\delta \mathrm{VC}_\mathcal{N}}$, and $\mathcal{N}_{\mathcal{D},\mathcal{W},\mathcal{S},\mathcal{B}}$ satisfying $C_8 \mathcal{D}\mathcal{S} \log \mathcal{S} \leq \mathrm{VC}_\mathcal{N} \leq C_9 \mathcal{D}\mathcal{S} \log \mathcal{S}$, see Bartlett et al. (2019). Then (22) follows from the above display with the selection of the network parameters of $\mathcal{M}_{\tilde{\mathcal{D}},\tilde{\mathcal{W}},\tilde{\mathcal{S}},\tilde{\mathcal{B}}}, \mathcal{N}_{\mathcal{D},\mathcal{W},\mathcal{S},\mathcal{B}}$ and with $\delta = \frac{1}{n}$. $\quad\square$

Finally, Theorem 4.3 is a direct consequence of (11) in Lemma B.1 and (16) in Lemma B.5. This completes the proof of Theorem 4.3. $\quad\square$

