# OpenReview forum: "Sufficient and Disentangled Representation Learning"
_ICLR.cc/2021/Conference — Reject_

### Official Review · AnonReviewer4 · 2020-10-26
**A GAN-like approach to representation learning that has little to do with disentangling**

**Rating:** 5
**Confidence:** 4

**Review:**

The authors present a new representation learning algorithm that trades off between a sufficiency condition (that is, the label should be independent of the input conditioned on the representation) and what they call a "disentangling" condition - that the representation vectors should be independent of one another and rotationally invariant. While the first condition has been used to define disentangled representations, the second is not standard. From the condition of rotational invariance, they require that the distribution over representations is isomorphic to a uniform Gaussian. They arrive at a loss with two terms, the first is a distance correlation between labels and representation, and the second is a divergence between the representation and a uniform Gaussian. In this sense, the regularization term looks quite similar to a VAE while the loss term looks quite similar to standard classification losses. The regularization is represented as a maximum over another loss, leading to a GAN-like coupled optimization problem.

The paper is written with quite a lot of complicated math. While I did not find anything wrong with the math, it did seem like at times it was meant more to obfuscate and impress and was often not really necessary. The experiments were almost entirely either comparisons against classic statistical methods (section 6.1 and 6.2, Table 1) or experiments on MNIST, FashionMNIST and CIFAR-10 (6.2, Table 3). Given the explosion of different deep learning methods for classification, representation learning, and disentangling in the last decade, it seems like the comparisons against classic statistical methods is missing the point - it seems highly likely that other deep learning methods could work just as well. In the experiments on MNIST, FashionMNIST and CIFAR-10, the authors set quite a high bar for themselves. These are 3 of the most over-studied datasets in all of machine learning - literally thousands, maybe tens of thousands of papers have been written on various deep learning algorithms applied to these datasets in the last several years. The experiments presented here do not seem to be a proper apples-to-apples comparison, however. A standard MLP is used for the SDRL (the method developed in this paper) while a DenseNet architecture trained with a classification loss is used for the "convolutional network" method. I was not able to find details on the dCorAE architecture. To properly reduce the number of possible confounding factors, the same architecture should be used with both the SDRL and the baseline methods, with different objective functions. Also, the classification numbers presented on CIFAR-10 do not seem to be state-of-the-art. According to Papers With Code, the DenseNet is able to achieve 96% accuracy on CIFAR-10, and even newer methods are able to reach 99% accuracy without additional data.

My biggest objection to the paper, however, is that it seems completely unrelated to disentangling. The only experiments presented are on visualization and classification - no results on standard disentangling tasks are presented. The rotational invariance condition seems completely against the grain of disentangling research. Despite disagreements over the exact definition of disentangling, there is at least broad agreement that a disentangled representation is one in which certain directions in latent space are privileged over others, and align with "true" latent factors in the world. While I disagree with the probabilistic definition of disentangling given in Locatello et al (2018), and prefer the geometric definition of Higgins et al (2018), I do agree with the point made in Locatello that, if different unique directions in latent space are not identifiable, then disentangling is not possible. Yet in this paper, the non-identifiability of different directions is given as a *necessary condition* for disentangling. This is the completely contrary to how the term is usually used. Even in the beta-VAE, which does have a rotationally-invariant loss function, the invariance is broken by requiring that the approximate posterior from the encoder has a diagonal covariance. Given this, I would recommend that the authors remove any reference to disentangling, and rewrite this purely as an alternative approach to supervised learning.

---

> ### Author Response · Authors · 2020-11-15
> **Response to AnonReviewer4**
>
> We’d like to thank you for taking the time to read our submission and for your comments. However, based on the comments we are afraid that you may have misunderstood the point of our work and did not provide a fair assessment of our contribution. We hope our responses below may help clarify the key points of our work and its contribution.
>
>
> •A challenge of supervised representation learning that distinguishes it from standard supervised learning is the difficulty in formulating a clear and simple objective function.  We are not aware of any simple empirical risk criterion for representation learning.  The criteria for characterizing conditional independence we know of are all unfortunately complicated and involve some math.
>
> The goal of our experiments is to demonstrate that the representations trained based on the proposed method perform well in terms of classification accuracy and prediction.  The results show that a simple classification model using the representations we trained performs better than or comparable with the best classification using deep CNN.  The comparison of classification experiments is apple-to-apple since the architecture of R_\theta and most hyperparameters were shared across all three methods – SDRL, CN, and dCorAE. The depth of the DenseNet in the paper with code (https://paperswithcode.com/paper/densely-connected-convolutional-networks) is about twice that of our model, so this comparison is indeed a bit unfair to our method. The network architecture of DenseNet we used is shown in the appendix. We are conducting further experiments based on your comments. At present, SDRL can achieve 97.35% accuracy on CIFAR-10.
>
> •We respectfully disagree. We use a narrow definition of disentangling: a representation is disentangled if the components of the representation are independent (see e.g., Achille and Soatto, 2018).  The rotation invariance (in distribution) condition is on top of the independence condition. So our definition is even stricter than the definition of disentangling as used in Achille and Soatto (2018).  For a given representation learned based on our method, different components (directions) are identifiable. What we mean by the representation being only identifiable up to orthogonal transformation is that if R is a learned representation, then any orthogonal transformation of R has exactly the same statistical property.
>
> The definition of Higgins et al. (2018) is based on the decomposition of a symmetry group into subgroups.  However,  it is not clear how to establish an objective function based on their definition to learn disentangled representations empirically.  We also do not see any contradiction between their definition and the definition we used.  Importantly, their definition does not imply that a disentangled representation must be unique since the decomposition of a symmetric group is generally not unique. In their grid world example (Section 5.3 Higgins et al., 2018), the symmetry group G has two decompositions: G = Gx X Gy X Gc, where Gx is the set of all translation transformations along the x-axis, Gy is the set of all translation transformations along the y-axis, and Gc is the set of all color transformations. Another decomposition is G = Gp X Gc, where Gp is the subgroup of all positional changes. These two decompositions lead to two different disentangled representations.
>
> In general, since we seek a representation in a space with a lower dimension than the dimension of the original data space, it is impossible to have uniqueness and identifiability in a nonparametric setting. Even if we restrict to linear representations with orthogonal directions, only space spanned by the linear representations is identifiable, but not the linear representation itself. This fact has long been established in the literature (See, for example, Cook, 2007; Fukumizu, Bach and Jordan, 2009 ).  Such representations satisfy the disentanglement definition of Higgins et al. (2018) by taking the symmetry group to be the group of orthogonal matrices.   In general, non-uniqueness and non-identifiability do not cause any problem for representation learning since the representations satisfying the required conditions are statistically equivalent and any one of them can be used for subsequent learning tasks.  While it would be nice if there existed a unique set of “true” latent factors in the world, we believe for most of the machine learning and statistical systems that have to be learned from empirical data, this is unfortunately not true.  There are multiple models and multiple representations that can lead to the same prediction accuracy (for a discussion about the multiplicity of good models, see e.g., Leo Breiman.  Statistical Modeling: The Two Cultures. Statistical Science, 16: 199-231, 2001).

---

### Official Review · AnonReviewer2 · 2020-10-27
**This framework is novel to me, but the paper needs to be revised**

**Rating:** 6
**Confidence:** 4

**Review:**

This paper proposes a new representation learning framework for supervised learning
In order to achieve good prediction accuracies while maintaining some desired properties.
The sufficiency and disentangled properties of the representation are formalized in
this paper to achieve this goal. Most results in the literature have focused on
the unsupervised setting, therefore this framework is novel to me.
Below I have a few comments to revise the paper.

To achieve sufficiency, this paper proposes to use the distance covariance defined in Section 4.
This seems to be related with statistical correlation,
can this be written out in the paper? As when I read the proof of
theorem 4.2, I do not know what is the \rho(R,R*). It seems not
to be defined in the paper.
Also is there any relation between V[z,y] with the sufficient statistics (Fisher 1922), as mentioned in the introduction?
From (5) I then understand that this covariance V replies on the metric on X and Y.
For multi-class classification problems, it is not always clear how to choose
a metric on Y. I recommend the authors to clarify this choice
at least in the numerical experiments.

In the statement of Theorem 4.3, should \lambda = O(1) be strictly positive? The objective (7) is formalized as a bi-level optimization problem, and the computation problem is addressed using recently proposed methods based on gradient flows. There is a typo 2 lines before Section 5, should the rate be (log n) n^{-2 / (2+d)} ?

Section 6 presents numerical results. I recommend to clarify what is Y in the classification problem
and how the metric is chosen. Figure 1 should be significantly improved. Figure 1 illustrates well that the learnt features are sufficient, but it is unclear whether the features are disentangled. To make the results more convincing, I suggest to plot the learned representation directly (maybe
in Appendix) to check whether it looks like an isotropic 2d Gaussian. Any statistical test of Gaussianity would be more convincing.

What is the distance correlation computed in Figure 2?
V[z,y] is called the distance covariance, so I guess it is something
else. Is the Table 3 showing the training or test accuracy?

Typo:
- in the definition of V[z,y] on page 4, the constant c_m should be c_q?
- 3 lines before (2), N_d (0, I_d) -> N (0, I_d)?
- The (a) and (e) in Figure 1 are not the learned features, but the original data.
- page 19: matric entropy -> metric entropy

---

> ### Author Response · Authors · 2020-11-15
> **Response to AnonReviewer 2**
>
> We’d like to thank you for the very detailed and constructive comments. Here are our point-by-point responses.
>
> • To achieve sufficiency, this paper proposes to use the distance covariance defined in Section 4.  This seems to be related with statistical correlation, can this be written out in the paper? As when I read the proof of  theorem 4.2, I do not know what is the \rho(R,R*). It seems not  to be defined in the paper.  Also is there any relation between V[z,y] with the sufficient statistics (Fisher 1922), as mentioned in the introduction?  From (5) I then understand that this covariance V replies on the metric on X and Y.  For multi-class classification problems, it is not always clear how to choose a metric on Y. I recommend the authors to clarify this choice at least in the numerical experiments.
>
> The population distance covariance is defined as the integral of the squared difference between the joint characteristic function and the product of the marginal characteristic functions. This is different from the usual ``statistical correlation’’ (e.g., Person’s correlation, Spearman’s rank correlation, or Kendal’s tau). As can be seen from the definition of the distance correlation, it is a measure of dependence (not just correlation) of two random vectors. In particular, the distance covariance is zero if and only the two random vectors are independent. The usual “statistical correlation” does not have such properties. The distance covariance does not have a direct connection with Fisher’s sufficient statistics. The connection is between the use of conditional independence to define a sufficient representation and Fisher’s sufficient statistics. Recall that for a given parametric model, a statistic is sufficient if the distribution of the data given this statistic is independent of the parameter, that is, a sufficient statistic contains all the information about the model parameter. The sufficient representation we seek to find is similar in the sense that given a sufficient representation, the input variable does contain any additional information about the response.
> The \rho(R, R*) is Pearson’s correlation coefficient.  For multiple class classification problems, we use vector representation for classes. Suppose there are k classes, then Y is a k-vector. We will make this clear in the revision.
>
> • In the statement of Theorem 4.3, should \lambda = O(1) be strictly positive? The objective (7) is formalized as a bi-level optimization problem, and the computation problem is addressed using recently proposed methods based on gradient flows. There is a typo 2 lines before Section 5, should the rate be (log n) n^{-2 / (2+d)} ?
>
> Yes, \lambda should be strictly positive. We will make this clear in the revision. Thanks for pointing out the typo. We will correct the typo in the revision.
>
> •  Section 6: clarify what is Y, how the metric is chosen, and plots of learned representations
>
> We will clarify the definition of Y in the numerical experiments and the metric used.  For the classification problem, Y is the label for image classification. For example, the label of MNIST is the corresponding number, and the label of CIFAR-10 is the type of object in the image.  In addition to the Gaussian distribution, we can choose a different distribution as our reference distribution for R*, such as a uniform distribution on the unit sphere. The point is that the components of this distribution are independent to meet the conditions of disentanglement. For the visualization of classification problems, the uniform spherical distribution is more suitable for displaying labeled low-dimensional representations than the Gaussian distribution. Therefore, in the experiment in Figure 1, we adopted the uniform spherical distribution. Since the dimension of the learned feature is 2, the visualization of uniform spherical distribution degenerates to the unit circle, which is what we show in Figure 1. We can see in Figure 1 that the points with different colors are well separated, which indicates that the features are well disentangled. We will also experiment with other ways for displaying the results as you suggested.
>
> •What is the distance correlation computed in Figure 2?
>
> The squared distance correlation \pho[z, y]^2 = V[z, y]^2/sqrt(V[z]^2 * V[y]^2), where V[z] , V[y] are the distance variances, V[z] = V[z, z], V[y] = V[y, y]. For more details, please see (Szekely et al., 2007). Table 3 is showing the test accuracy.
>
> Typo:
>
> •	in the definition of V[z,y] on page 4, the constant c_m should be c_q?
>
> We will correct this typo in the revision.
>
> •	3 lines before (2), N_d (0, I_d) -> N (0, I_d)?
>
> We will make the change in the revision
>
> •	The (a) and (e) in Figure 1 are not the learned features, but the original data.
>
> We will make the corrections in the revision.
>
> •	page 19: matric entropy -> metric entropy
>
> We will correct this typo in the revision.

---

### Official Review · AnonReviewer1 · 2020-10-28
**A solid work that aims to learn sufficient and disentangled representations, with strong motivations, sound theoretical justification and extensive empirical validations.**

**Rating:** 7
**Confidence:** 4

**Review:**

This work proposes a method, SDRL, for learning sufficient and disentangled representations, with the additional property that the representations should also be rotation-invariant. Together with the disengtangled property, the ration-invariant property specifies the distribution of representations to be isotropic Gaussians. On the other hand, the repressentations are required to be sufficient for predicting the target labels. These two goals motivate the Lagrangian formulation of the objective function, based on which the authors apply two different estimators for these two goals. Experiments are extensive, with a good mixture of synthetic and real-world data. I particularly like the visualization of the learned representations on the synthetic data, which well corroborates the theoretical claims in the first part of the paper.

Overall, I think the paper has a sound logic and clear presentation. The only downside is that the novelty is a bit limited: both the sufficiency criterion (see Tishby et al. 2015) and the disentangled criterion (mutual independency) have well been studied in the literature, despite using other divergence measures, e.g., mutual information and KL divergence. That being said, I haven't seen similar work using exactly the same divergences as the ones in this paper, e.g., distance correlation and f-divergence in this context.

A few more minor comments and questions:
-   I could understand representations that are rotation-invariant are desirable in vision tasks, but I am wondering is this requirement too strong for other applications, e.g., in language and speech? Other than the technical convenience brought by this property, is there any other reason that could motivate this criterion?

-   In Eq. (2), shouldn't the domain of g be R^p instead of R^d? Otherwise the intersection of M and F would be empty.

-   If my understanding about Lemma 2.1 is correct, it is the pushforward of \mu under T to be an isotropic Gaussian distribution, right? The original Z = g(X) is only sufficient, but not disintangled nor rotation-invariant. If this is the case, then I am not sure I totally agree with the second comment after Theorem 4.2, where the authors remark that the discriminator network is the pushforward function T. My understanding is that the representer network R is used to approximate the composite of T and g, so that R_\sharp \mu_X = R^* in Eq. (3), while the discriminator network D is only served as the witness function in the definition of f-divergence in Eq. (6). Please clarify this point, thanks!

-   The line above Eq. (9), typo: no \lambda is needed before the V term.

-   IMO the second assumption of Theorem 4.3 is quite strong: it requires the density to be lower bounded by a positive constant c_1, which does not even hold for Gaussians. On the other hand, the compactness of the domain helps to reconcile this assumption, and I understand that this is a standard assumption in the analysis of nonparametric density estimation/regression.

---

> ### Author Response · Authors · 2020-11-15
> **Response to AnonReviewer1**
>
> We’d like to thank you for the very detailed and constructive comments. Here are our point-by-point responses.
>
> Thank you for noticing the merits of our paper.
>
> • I could understand representations that are rotation-invariant are desirable in vision tasks, but I am wondering is this requirement too strong for other applications, e.g., in language and speech? Other than the technical convenience brought by this property, is there any other reason that could motivate this criterion?
>
> Indeed, for language and speech data, rotation invariance in distribution may not be desirable. Some other reference distributions other than normal may be more appropriate. In the proposed method, we can also push the distribution of the representation to other distribution such as uniform.  We can also think of pushing the distribution of the representation to the Gaussian distribution as a form of regularization. We will add these points to the revision.
>
> •	In Eq. (2), shouldn't the domain of g be R^p instead of R^d? Otherwise the intersection of M and F would be empty.
>
> Yes, the domain of g should be R^p in Eq. (2). Thanks for catching this error.
>
> •	If my understanding about Lemma 2.1 is correct, it is the pushforward of \mu under T to be an isotropic Gaussian distribution, right? The original Z = g(X) is only sufficient, but not disintangled nor rotation-invariant. If this is the case, then I am not sure I totally agree with the second comment after Theorem 4.2, where the authors remark that the discriminator network is the pushforward function T. My understanding is that the representer network R is used to approximate the composite of T and g, so that R_\sharp \mu_X = R^* in Eq. (3), while the discriminator network D is only served as the witness function in the definition of f-divergence in Eq. (6). Please clarify this point, thanks!
>
> Yes, you are right. That comment is misleading. The discriminator network D only serves as a witness function as in GANs. Many thanks for pointing this out.
>
> •	The line above Eq. (9), typo: no \lambda is needed before the V term.
>
> Yes, this is a typo. We will remove \lambda in the revision.
>
> •IMO the second assumption of Theorem 4.3 is quite strong: it requires the density to be lower bounded by a positive constant c_1, which does not even hold for Gaussians. On the other hand, the compactness of the domain helps to reconcile this assumption, and I understand that this is a standard assumption in the analysis of nonparametric density estimation/regression.
>
> We agree this is a strong assumption. We now can weaken this assumption by only assuming the density to be lower bounded by a positive constant on a sufficiently large compact set. This will include Gaussian densities. We can use a truncation argument so that the original proof will only need minor modifications. We will make the changes in the revision.

---

### Official Review · AnonReviewer3 · 2020-10-29
**Review for "Sufficient and Disentangled Representation Learning"**

**Rating:** 4
**Confidence:** 5

**Review:**

This paper introduces SDRL, a representation learning algorithm for supervised learning.
SDRL aims to enforce three constraints to the learned representation g(x) :
1) That all the information from X about Y is preserved: Y independent to x given g(x)
2) That it's distributionally rotation invariant: Law(O g(x)) = Law (g(x))
3) That all it's coordinates are independent: Law(g(x)_i) = Law(g(x)_j) for all i, j

Since 2 and 3 are equivalent to asking that g(x) is Gaussian, they look for a pushforward map g such that g(x) is Gaussian.
In particular, they pick the optimal transport one given by Brenier's theorem.

I think the paper needs more work in the following on 4 very important aspects:

A) The paper is not very well motivated in my opinion. Why do we want to learn a representation with statistically independent and rotation invariant coordinates? I know many works already try to achieve this goal, but this paper doesn't do a good job at motivating it. It merely cites a bunch of other papers that do so, but reading the paper I am left with the impression of "we want to come up with a representation with properties 1, 2, 3. Why? Ask other people". It doesn't feel like I'm given a sufficient explanation of why I should be reading this paper and the reality is that if I wasn't asked to review it I would have left it in the pile of papers in my desk half way through the introduction. This is very concretely seen in the sentence "We also require that the representation is rotation invariant in distribution. Such an invariance property is an important characteristic in many visual and morphological tasks \cite{}". This is a third of the problem you're trying to solve, it is a bit crazy that all the motivation it has is a citation to two papers that are not mentioned anywhere else in the paper.

B) The method is very complicated. Why use the distance covariance rather than the empirical risk to assure that the information from X about Y is preserved in G(X)? I know the authors have several choices for V, but the choice they pick is quite particular and a complicated one at it. Without much motivation or experiments ablating this choice it's hard to justify it. It's hard to convince the reader to bother like this.

C) The learning bounds are pretty irrelevant. the n^-{2/(2+d)} bound, while minimax optimal, is useless in high dimensions (high meaning d > 10, which is usually what we are interested in representation learning). While minimax optimal ***in the distributionally worst case***, it is irrelevant in practice (and it is not an advance in any theoretical field). It's simply taking extra space in the paper, it really brings nothing to the table in my opinion.

D) The experiments are extremely underwhelming. For such a complicated method does not bring any significant benefit in the real classification datasets, and a very minor one in the regression ones. In regression, furthermore, an important baseline is missing: the last layer of an NN trained with a regression and / or ordinal regression loss. It seems unfair to compete with the NN inductive bias in the author's algorithm vs linear algorithms.

Overall, the authors take us through an interesting but overly complicated mathematical journey with a poor motivation, irrelevant learning bounds, and no empirical evidence that the journey was useful whatsoever.

---

> ### Author Response · Authors · 2020-11-15
> **Response to AnonReviewer3**
>
> We’d like to thank you for taking the time to read our submission and for your comments. However, based on the comments we are afraid you may have misunderstood the key points of our paper and did not provide a fair assessment of our contribution. We hope our responses below may help clarify the key points of our work and its contribution.
> Our point-by-point responses are as follows.
>
> First of all, your following description of independence among the coordinates is totally wrong:
>
> 3. That all it's coordinates are independent: Law(g(x)_i) = Law(g(x)_j) for all i, j
>
> A)	Our work builds on the existing work, so we believe it is important to credit the researchers for their work. Sufficiency is a basic statistical property that a representation should have. This is a well-established principle in statistics.  Disentanglement is a condition that has long been recognized as an important desired condition for representation learning (see, for example, Bengio (2013) and Achille and Soatto (2018)). The rotation invariance property is motivated by image data and morphological data. Indeed, we should have better motivated this in addition to what we have learned from the existing literature.  Also, we proved that we can always transform the representation to have independent components and be invariant in distribution while preserving sufficiency (Lemma 2.1). So, there is no loss of information in a statistical sense to impose invariance on the representation. In the revision, we will better motivate the study and explain in more detail why we proposed the method to learn sufficient and disentangled representations.
>
> B)	We are not aware of any simple empirical risk criterion for representation learning in a nonparametric setting. To the best of our knowledge, we don’t think such a criterion has been proposed in the literature. A challenge of supervised representation learning that distinguishes it from standard supervised learning tasks is the difficulty in formulating a clear and simple objective function. In classification, the objective is clear, which is to minimize the number of misclassifications; in regression, a least-squares criterion is usually used.  In representation learning, the objective is different from the ultimate objective, which is typically learning a classifier or a regression function for prediction. How to establish a good and simple criterion for supervised representation learning has remained an open question (Bengio, 2013).
>
> In our work, we develop a criterion for supervised representation learning by formulating the problem as that of finding a representation that satisfies the conditional independence condition, and then regularizing it by requiring its components to be independent and rotation invariant (in distribution).  So we need a measure of conditional independence. Distance covariance is actually one of the statistically and computationally simplest ones for characterizing conditional independence. We mentioned two other possibilities：mutual information and kernel covariance operator, but they are not any simpler than distance covariance. Specifically, mutual information involves the estimation of density ratios, in addition to the representation function. Nonparametric density-ratio estimation is a challenging problem in itself in high-dimensional settings. The computation of the empirical kernel covariance operator is expensive if not infeasible.  We would be very grateful if you could let us know a simple empirical risk function for characterizing conditional independence and disentanglement and hence can be used for learning a sufficient and disentangled representation nonparametrically.
>
> C)  We believe that the learning bounds provide strong support for the proposed method. We agree this is not directly related to training and application. However, this result shows that the proposed method at least is doing the right thing under the conditions provided.
>
> D)  The goal of our experiments is to demonstrate that the representations trained based on the proposed method perform well. Our proposed method is not trying to learn a classifier or a regression function directly, but rather to learn a representation that preserves all the information. So our experiments are designed to evaluate the performance of simply classification and regression methods using the representations we learned as input. The results demonstrate that a simple classification model using the representations we trained performs better than or comparably with the best classification method using deep CNN.
> Thank you for your suggestions on the setting of regression problems. We will add experimental comparisons of NN-based neural network models in the revision according to your suggestions.

---

### Decision · Program_Chairs · 2021-01-07
**Final Decision**

**Decision:**

Reject

**Comment:**

This paper aims to present a new representation learning framework for supervised learning based on finding a representation such that the input is conditionally independent given the representation, the components of the representations are independent and the representation is rotation-invariant. While there were both positive and negative assessments of this paper by the reviewers, there are 3 major concerns that lead me to recommend rejecting this paper:
1. Most importantly, experiments do not seem to be conclusive as they do not properly ablate the specific aspects of this method. More specifically, the authors compare their deep learning based approach with non-deep learning approaches but do not compare against deep learning baselines. This makes it impossible to assess the merit of the proposed approach (which also appears to be complicated) over much simpler baselines.
2. The required properties of the representations do not seem to be properly motivated.
3. The paper refers to their produced representations as disentangled representations. As pointed out by AnonReviewer4, this appears not to be consistent with prior uses of that word in the community.